# Prolonged survival of venereal *Tritrichomonas foetus* parasite in the gastrointestinal tract, bovine fecal extract, and water

Cristian I. Martínez,[1,2] Lucrecia S. Iriarte,[1,2] Nehuen Salas,[1,2] Andrés M. Alonso,[2,3] Cesar I. Pruzzo,[4] Tuanne dos Santos Melo,[5] Antonio Pereira-Neves,[5] Natalia de Miguel,[1,2] Veronica M. Coceres[1,2]

**ABSTRACT** Bovine tritrichomonosis is a venereal disease that causes economic losses around the world. Over the last 100 years, the life cycle of the protozoan *Tritrichomonas foetus* in bovines has only considered venereal transmission, almost exclusively through natural mating from an infected animal to a healthy animal. Here, we provide the first direct evidence that *T. foetus* can survive the passage through the gastrointestinal tract in bovines. We demonstrated that the parasite can be discharged by feces in orally infected animals and contaminate the cow's reproductive tract. We showed that the parasite is capable of surviving at pH conditions usually found along the bovine gastrointestinal tract, and our findings suggest that the parasites could survive in bovine feces and water for several days, probably through the formation of resistant pseudocysts or cyst-like structures. Taken together, our results demonstrate that *T. foetus* is more resistant to adverse external conditions than previously considered, which could be relevant for the life cycle of this parasite in bovines.

**IMPORTANCE** Nowadays, the routine herd diagnosis is usually performed exclusively on bulls, as they remain permanently infected, and prevention and control of *Tritrichomonas foetus* transmission are based on identifying infected animals and culling practices. The existence of other forms of transmission and the possible role of pseudocysts or cyst-like structures as resistant forms requires rethinking the current management and control of this parasitic disease in the future in some livestock regions of the world.

**KEYWORDS** *Tritrichomonas foetus*, life cycle, bovines

*T*ritrichomonas *foetus* (family Tritrichomonadidae) is a widespread anaerobic pathogen that colonizes the reproductive tract of cattle and the large intestine of cats, leading to bovine and feline tritrichomonosis, respectively (1). The life cycle of *T. foetus* is considered monoxenous, meaning that development is restricted to a single host species. *T. foetus* has two known cellular forms: trophozoites and pseudocysts. Such pseudocysts have been described as trophozoites that adopt a spherical shape, carry out nuclear division, form multinucleated cells, and internalize their flagella under unfavorable environmental conditions (2, 3), although their role in the Tritrichomadidae life cycle is still unknown.

Bovine tritrichomonosis is a venereal disease that causes economic losses in beef and dairy farming due to early embryonic deaths, abortion, and the consequent reduction in the conception rate and the number of calves born in the herd. Moreover, added to these economic losses are expenses associated with the financial costs of testing bulls, replacing infected bovines, and veterinary expenses (4–6). The disease is widespread around the world, especially in Asia, Australia, South Africa, and South America, where natural service by bulls is used as a major means of breeding (4, 5, 7). Over the last 100 years, the life cycle of *T. foetus* in bovines (*Bos taurus taurus* and *Bos taurus indicus*)

Address correspondence to Veronica M. Coceres, coceres@intech.gov.ar.

Cristian I. Martínez and Lucrecia S. Iriarte contributed equally to this article. Author order was determined by drawing straws.

The authors declare no conflict of interest.

has only considered the venereal transmission almost exclusively through natural mating from an infected animal to a healthy animal (8). The bulls become infected during mating with infected cows, remaining as asymptomatic carriers and being responsible for the disease spreading rapidly through the herd. Taking this into account, the current control programs are based on the testing and exclusion of individual bulls due to a lack of effective treatment regimens. Being the standard diagnostic methods are microscopic observation (with or without prior cultivation on media) and molecular tests [polymerase chain reaction (PCR)] (9, 10). Besides, the parasite can be transmitted mechanically during the practice of artificial insemination or vaginal examination if contaminated material is used (8). Considering the different routes of infection described in bovines, the apparent presence of *T. foetus* in samples from virgin bulls (11) or the presence of bull-negative samples and cow-positive samples in the same bovine herds reported by veterinarians (especially in those regions where the extensive system predominates) are clearly situations difficult to explain based on the current knowledge of the *T. foetus* life cycle.

Trichomonads represent a group of flagellated protozoa adapted for living in anaerobic and microaerobic environments, with a great capacity for different context adaptations. Most of these organisms inhabit the intestines of diverse hosts (vertebrates and invertebrates). However, only a few of these organisms have been shown to have pathogenic potential. Among the Trichomonadidae family, *Trichomonas gallinae* is a parasite that infects the upper digestive tract (columbiformes, falconiforms, raptors, and passeriforms); *Tetratrichomonas gallinarum* is an intestinal parasite (galliform and anseriform birds); and *Trichomonas tenax* is a commensal of the oral cavity (humans, dogs, and cats). *Trichomitus batrachorum* is an amphibian intestinal parasite, and *Pentatrichomonas hominis* is a commensal protozoan (mammal intestine). In this sense, *T. foetus* is also a symbiont found in the intestines and stomachs of pigs. Moreover, *T. foetus* and *P. hominis* were recently detected in the feces of dogs with diarrhea (12, 13), and *T. foetus* spread in cats involves fecal-oral transmission (14, 15). In addition to the gastrointestinal tract infection in cats, Dahlgren et al. reported the first case of *T. foetus* in the uterus of a cat with pyometra (16).

On the other hand, the bovine gastrointestinal tract hosts other commensal trichomonads (e.g., species of *Pentatrichomonas*, *Tetratrichomonas*, and *Pseudotrichomonas*), which in some cases have been isolated from the reproductive tract of cattle, especially in areas where natural breeding of beef cattle is widely practiced (9, 17–20). In this context, the presence of these trichomonads in samples collected for *T. foetus* testing thus represents a potential problem during routine screening for bovine tritrichomonosis. Although in different countries, artificial insemination and control programs have greatly reduced the apparition of trichomonads of gastrointestinal origin and *T. foetus* isolated from the bull's preputial cavity (21, 22).

This report provides the first direct evidence that *T. foetus* can survive passage through the gastrointestinal tract in bovines. Moreover, we demonstrated that the parasite can be discharged by feces in orally infected animals and contaminate the cow's reproductive tract. On the other hand, we also revealed the presence of *T. foetus* in the feces of naturally infected animals. We showed that the parasite is capable of surviving at pH conditions usually found along the bovine gastrointestinal tract and can persist in bovine feces as well as in water for several days. Finally, we have demonstrated that *T. foetus* forms cyst-like structures under certain conditions of stress. Taken together, our results demonstrate that *T. foetus* survives in the gastrointestinal tract, bovine fecal extract, and water, which could be relevant for the life cycle of this parasite in bovines.

## RESULTS

### *Tritrichomonas foetus* can survive throughout the bovine gastrointestinal tract and contaminate the cow's reproductive tracts

It is widely accepted that bovine tritrichomonosis is a venereal disease. However, some unexplained circumstances, such as the presence of *T. foetus* in samples from virgin bulls or the presence of bulls negative for *T. foetus* cohabiting with cows positive for *T. foetus* in the same herd, have been reported by veterinarians, especially in areas where natural breeding of beef cattle is widely practiced (11). Additionally, even though *T. foetus* is a gastrointestinal parasite in cats, Dahlgren et al. have reported the first case of *T. foetus* in the uterus of a cat with diarrhea, suggesting that vaginal contamination might occurr during soft or liquid feces elimination (16). Based on these observations, we hypothesized that a similar mechanism of transmission might be occurring in bovines. To evaluate this, we first investigated whether bovines were capable of excreting viable *T. foetus* parasites after ingesting trophozoites. To this end, *T. foetus* trophozoites were administered orally to two cows for 5 consecutive days. As 72 h is the typical gastrointestinal transit time of bovines, cow feces samples were collected per rectum from days 3 to 10 (Fig. 1A). Moreover, vaginal lavage samples were obtained on day 7 (Fig. 1A), and day 0 as a control (negative). Initially, the presence of rounded parasites (classical shape of pseudocyst-like cells) (Fig. 1B), positive for WGA-FITC staining (Fig. 1C), were detected in the fresh feces samples since day 1 of these samples' collection. Then, the feces and vaginal mucus samples were maintained in trypticase-yeast extract-maltose (TYM) media culture and analyzed daily to evaluate the presence of motile *T. foetus* trophozoites by light microscope. After 24 h incubation in TYM media, motile organisms were detected in all the tubes inoculated with feces samples (Fig. 2A). The presence of motile *T. foetus* trophozoites and spherical pseudocyst-like cells were also detected in vaginal mucus samples (Fig. 2B). To demonstrate that the parasites present in the feces and mucus samples of the two cows (cow 1 and cow 2) were *T. foetus,* we amplified a 347-bp DNA fragment using the TFR3/TFR4 primers that specifically amplified the *T. foetus* RNA 5.8S gene (Fig. 2C). These results demonstrate that *T. foetus* can survive in

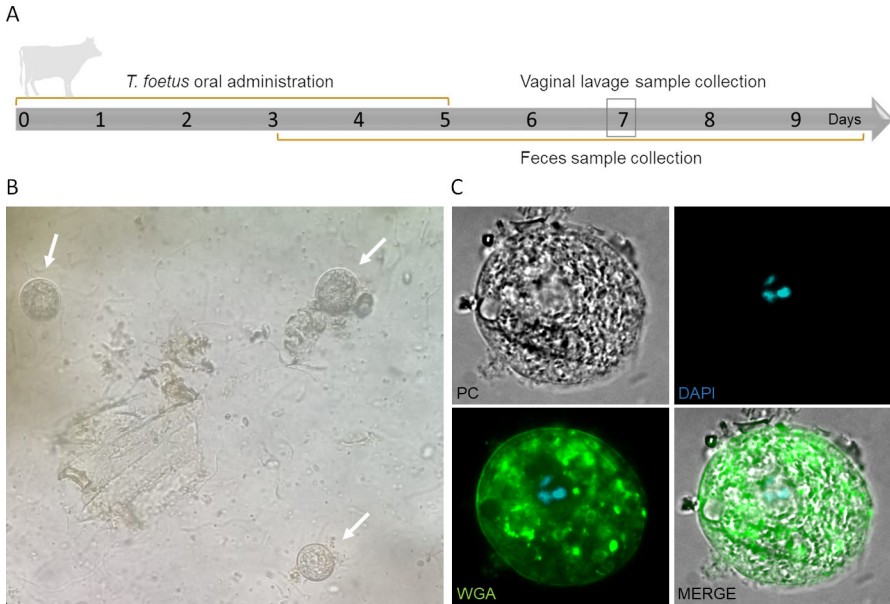

**FIG 1** Orally administered *T. foetus* survive through the bovine gastrointestinal tract and are discharged by feces. (A) Timeline, experimental design, and sample collection to evaluate *T. foetus* survival through the bovine gastrointestinal tract. (B) A representative optical microscopy image of spherical structures (white arrows) present in fresh feces samples after 4 days of *T. foetus* K strain oral administration. (C) A representative image analyzed by epifluorescence microscopy of pseudocyst-like structures stained with WGA in fresh feces samples after 4 days of *T. foetus* oral administration.

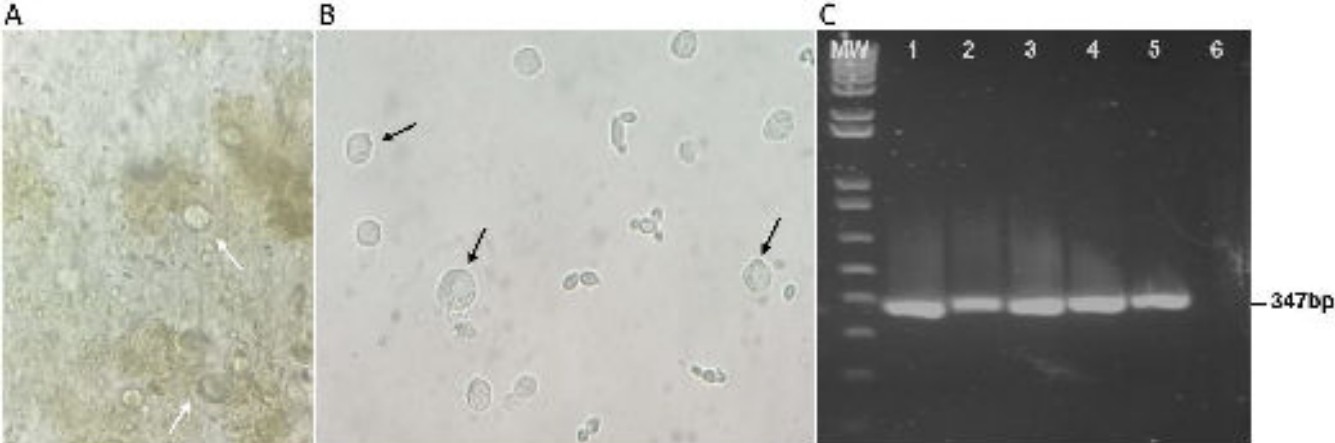

**FIG 2** Identification of *T. foetus* in vaginal mucus and feces. (A) A representative optical microscopy image of trophozoites and spherical structures present in TYM media inoculated with feces samples. The white arrows indicate the trophozoites and spherical structures. (B) A representative optical microscopy image of trophozoites and spherical structures present in vaginal mucus samples. The black arrows indicate the trophozoites and spherical structures. (C) Agarose gel electrophoresis of PCR products amplified with primers specific for a 347-bp segment of the *T. foetus* RNA 5.8S gene. MW (1 Kb Plus DNA ladder); lane 1, cow feces sample 1 (cow 1); lane 2, cow feces sample 2 (cow 2); lane 3, cow vaginal mucus 1 (cow 1); lane 4, cow vaginal mucus 2 (cow 2); lane 5, positive control (*T. foetus* K strain), and lane 6 sterile distilled water (negative control).

the bovine digestive tract and may contaminate the cows' mucosal vagina due to its elimination through feces.

## *Tritrichomonas foetus* is found in the feces of naturally infected bovines

To confirm a possible new route of transmission for *T. foetus*, we analyzed feces samples obtained from a herd of extensive livestock production, whereas bulls have been diagnosed positive for *T. foetus* using preputial samples by routine diagnostic methods in a reference laboratory (observation under the microscope and PCR). We evaluated the presence of *T. foetus* in feces samples from five bulls and five cows as well as five vaginal mucus samples from "empty cows" (non-pregnant cows). The obtained samples were incubated in TYM culture media at 37°C and the presence of motile trophozoites in each tube was examined by optical microscopy for 15 consecutive days. Motile organisms were detected in two bovine feces samples (bull number 3 and cow number 2) as well as in vaginal mucus samples of cow's numbers 2, 3, and 5 (Fig. 3A). These organisms were identified as *T. foetus* based on classical morphological features and the typical rolling-type motion of live flagellates using light microscopy (data not shown) and PCR using specific primers (Fig. 3B). Taking into account previous reports suggesting that false-positive results may occur in PCR tests using samples isolated from both bulls and/or female cattle (23, 24), we confirmed the presence of *T. foetus* by scanning electron microscopy (SEM). As can be observed in Fig. 3C, classical parasite ultrastructures, such as a piriform cell body with three anterior flagella and a single recurrent flagellum associated with an undulating membrane and free-ended at the posterior region, were observed (25, 26). Moreover, we sequenced the obtained PCR fragments (Fig. 4A) and found the sequences to be homologous (Fig. 4B).

Our results demonstrate the presence of *T. foetus* in feces of naturally infected bovines.

## *Tritrichomonas foetus* can survive in bovine feces for several days

Considering that it is plausible that *T. foetus* is transmitted by a fecal-oral route from an infected bovine to an uninfected one, trophozoites would need to survive in different environmental conditions to be transmitted. Specifically, the parasite needs to overcome the environmental conditions (temperature and lack of nutrients) during the period

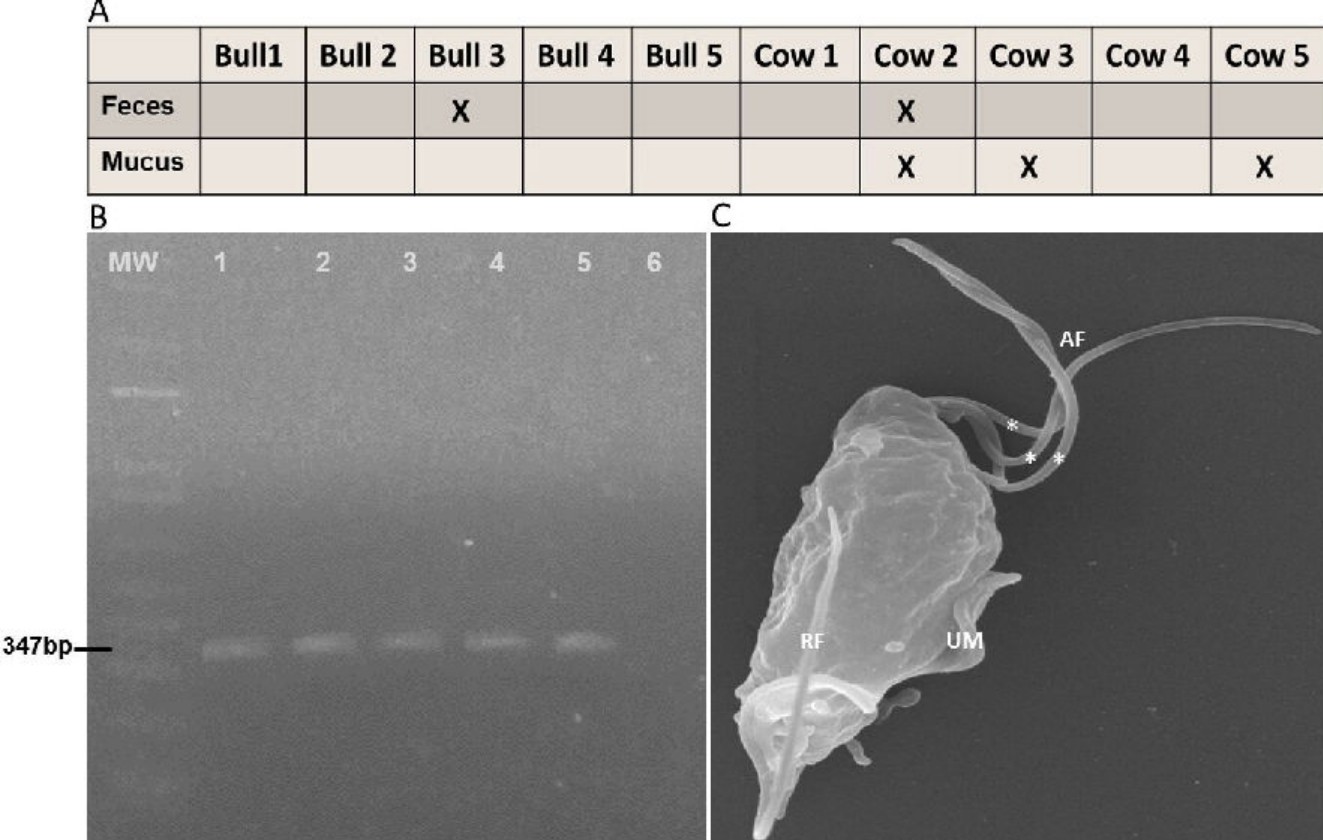

FIG 3 *T. foetus* detection in vaginal mucus and feces of naturally infected animals. (A) Graph 1: A resume of PCR analysis results for feces samples and vaginal mucus samples (X indicates the positive samples). (B) Agarose gel electrophoresis of PCR products amplified with primers specific for a 347-bp segment of the *T. foetus* RNA 5.8S gene. MW (1 Kb Plus DNA ladder); lane 1, cow feces sample 2; lane 2, cow vaginal mucus 2; lane 3, cow vaginal mucus 3; lane 4, cow vaginal mucus 5; lane 5, bull feces sample 3; and lane 6, sterile distilled water (negative control). (C) Representative SEM image of *T. foetus* isolated from bovine naturally infected. Parasite exhibits three anterior flagella (*) and one recurrent flagellum with a distal free end (RF) and associated with an undulating membrane (UM).

between being discharged from one host and being ingested by the next, as well as to survive in the hostile gastric niche (low pH) of the new host when ingested. Hence, we analyzed the parasite survival capacity under different stress conditions. Initially, we examined the ability to grow/survive of *T. foetus* in bovine feces. As that fecal samples contain a lot of coarse material (vegetable fibers, microorganisms, among others) that makes it difficult to visualize the presence of trophozoites in the samples, we used an extract of bovine feces to perform these assays. Thus, $1 \times 10^5$ parasites were inoculated in bovine feces extract (supernatant obtained from samples of feces mixed with phosphate-buffered saline (PBS) in a 1:1 ratio and centrifuged for 10 min at 3000 × *g*), incubated at room temperature (RT), and parasite counts were recorded using a Neubauer hemocytometer. We detected motile parasites in extracts from bovine feces for up to 96 h (Fig. 5A). Next, to analyze the speed at which microorganisms multiply during specific intervals, we estimated the growth rate of parasites in standard culture media and compared it to the rate of parasite growth incubated in bovine feces extract. The results showed that although the growth rate was greater in the complete TYM culture medium (6.66%), the parasites were able to grow in bovine feces extract (growth rate: 1.48%) (Fig. 5B).

We observed cells with spherical shapes in bovine feces cultures (Fig. 5C). It has been reported that, under unfavorable environmental conditions such as nutritional stress and low temperatures (4°C), the piriform trophozoites (a polar and flagellated cell) can adopt a spherical shape with internalized flagella known as "pseudocyst" or endoflagellar form

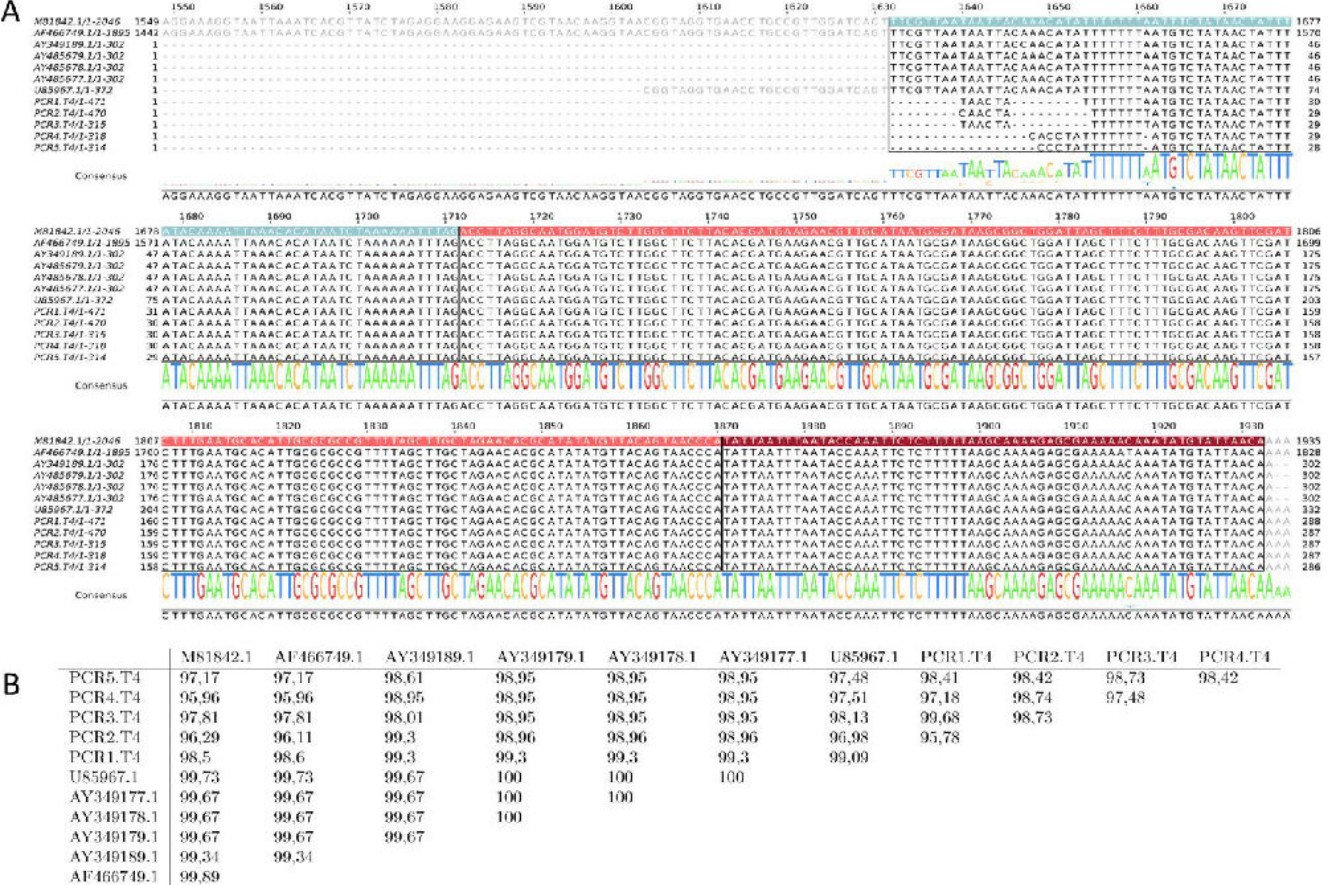

**FIG 4** Sequence analysis. (A) A multiple sequence alignment of DNA fragments corresponding to the complete ITS-1/5.8S/ITS-2 region from *T. foetus*. Highlighted regions correspond to ITS-1 (sky blue), the 5.8S gene (pink), and ITS-2 (red). Accession numbers are related to the strains listed below: M81842.1: CROP-1; AF466749.1: NCSU Tfs-1; AY349189.1: KV1; AY485679.1: PAL; AY485678.1: VAL; AY485677.1: K; U85967.1: *Tritrichomonas suis*. Codes for the sequences obtained in this work are listed below: PCR1 (cow feces sample 2), PCR2 (cow vaginal mucus 2), PCR3 (cow vaginal mucus 3), PCR4 (cow vaginal mucus 5), and PCR5 (bull feces sample 3). Corresponding percentage of identities is shown in Fig. 4B.

(2). To evaluate if the structures observed in the bovine feces extract were pseudocysts or possible cyst-like structures, parasites grown in bovine feces extract for 10 d were fixed and stained with Calcofluor White Stain (CFW), a non-specific fluorochrome that binds to the surface of pseudocysts and cyst-like structures of different organisms (27–30). As can be observed in Fig. 5C, our results demonstrate the presence of CFW + structures in analyzed samples.

Taking into account that Iriarte et al. (27) showed that the nuclei number in *T. foetus* was variable during deprivation or depletion of nutrients (stress), here we determined the number of nuclei per cell by fluorescence microscopy using DAPI staining. As shown in Fig. 5D, we found 34.5%, 34.5%, and 31% of parasites with one nucleus, or 1N, two nuclei (2N), and more than two nuclei (>2N) when parasites (K strain) are grown in the logarithmic growth phase in standard TYM culture media (control). Alternatively, 100% of parasites with one nucleus (1N) are detected when parasites are grown for 10 days in extract feces (EF), indicating that these growth conditions enrich a population of parasites with one nucleus. Interestingly, these nuclei exhibit a larger size (Fig. 5E, right panel) than parasite nuclei cultivated in conventional TYM culture media (Fig. 5E, left panel). Previously, the presence of parasites with a larger nucleus has been observed in *T. foetus* that endoreplicate their DNA during nutritional stress. In this context, it has been demonstrated that these polyploid parasites, when the nutrient supply is restored, undergo karyokinesis to generate several nuclei and then multiple parasites (27).

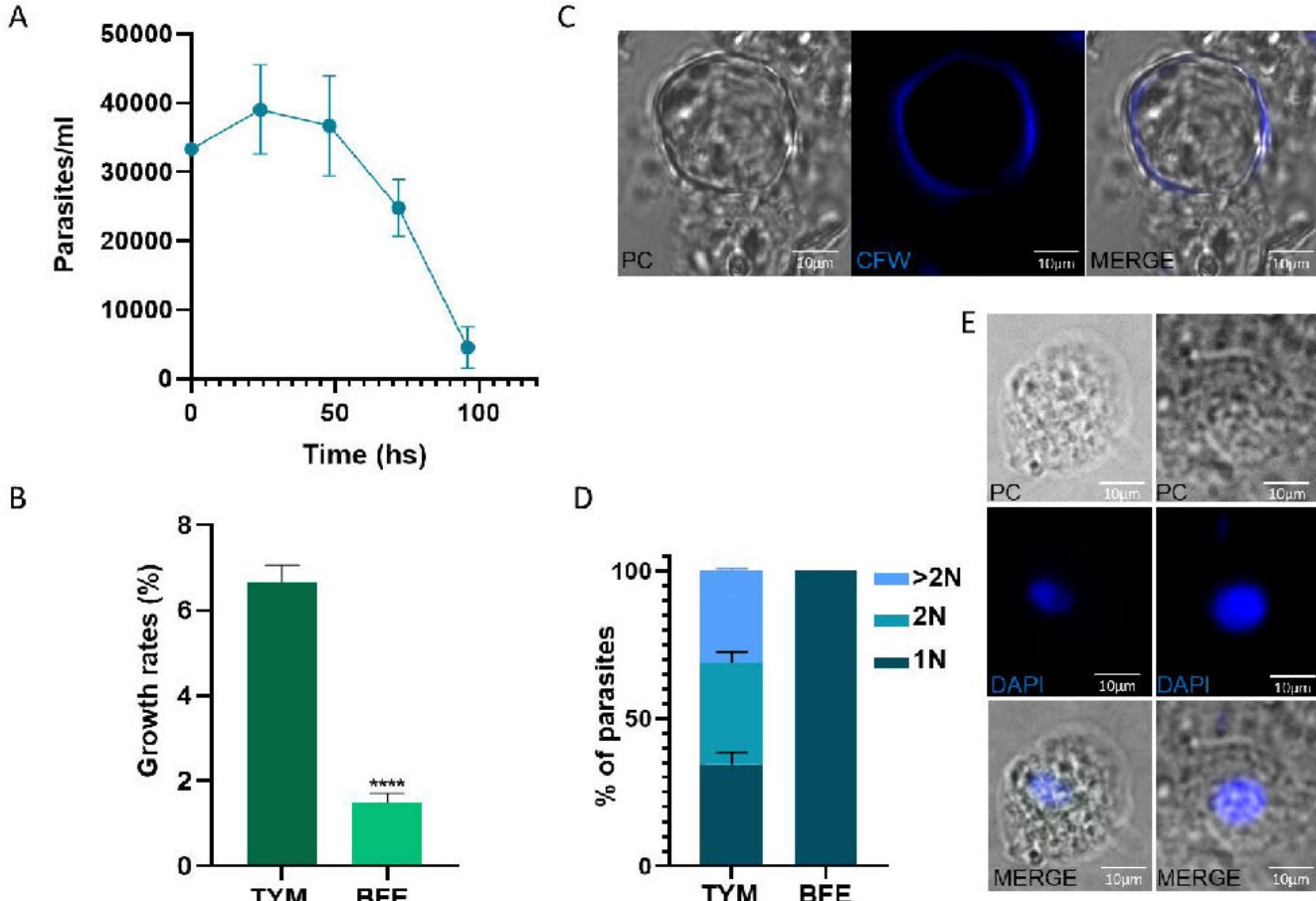

**FIG 5** Parasite growth in an extract from bovine feces. (A) Analysis of the *T. foetus* kinetic growth curve. Replication curves of parasites' growth in extracts of bovine feces were compared. Parasite counts were collected at the indicated times on the *X*-axis. Each point represents the average parasite concentration in 10 different tubes containing different feces samples. (B) Growth rates of parasites grown in standard culture media (TYM) were compared to parasites grown in an extract of bovine feces (BFE). Error bars represent standard error, and asterisks denote statistically significant differences determined by t-Student ($P <$ 0.05). (C) An example of a pseudocyst stained with Calcofluor white (CFW) in a *T. foetus* grown in bovine feces extracts for 10 days. The samples were analyzed by epifluorescence microscopy. (D) Quantification of the number of nuclei per parasite. Fifty parasites from each population were counted in triplicate in three independent experiments. The percentages of parasites with one nucleus (1N), two nuclei (2N), or more than two nuclei (>2N) are shown. *T. foetus* grown under standard culture media (TYM) and parasites grown in extracts from bovine feces (BFE) were compared. The results represent the average of three independent experiments and the standard error (SE). (E) Comparative images of *T. foetus* parasites with one nucleus grown in standard culture media (TYM) (left panel) and parasites grown in extracts from bovine feces (BFE) (right panel). Nuclei were stained with DAPI (blue). PC, phase-contrast.

These findings suggest that the parasites could survive in bovine feces for several days, probably through the formation of resistant mononucleate pseudocysts or cyst-like structures.

## *T. foetus* is capable of surviving in water for several days

Taking into account that bovines need to drink water 10–15 times a day for half a minute (around 10 L each time), they should have access to water in paddocks, and usually feed-out areas for several animals are available. Based on this, we hypothesized that water could be a vehicle for parasite transmission when ingested. To analyze if *T. foetus* was capable of surviving in water, we analyzed the viability by propidium iodide staining when incubated in water for up to 120 h (Fig. 6A). We concluded that parasites did not show viability differences among themselves when incubated in water for 48, 72, 96, and 120 h (Fig. 6B).

Taking into account that the incubation in water induced the transformation of pyriform trophozoites into spherical shapes as pseudocyst structures and that this

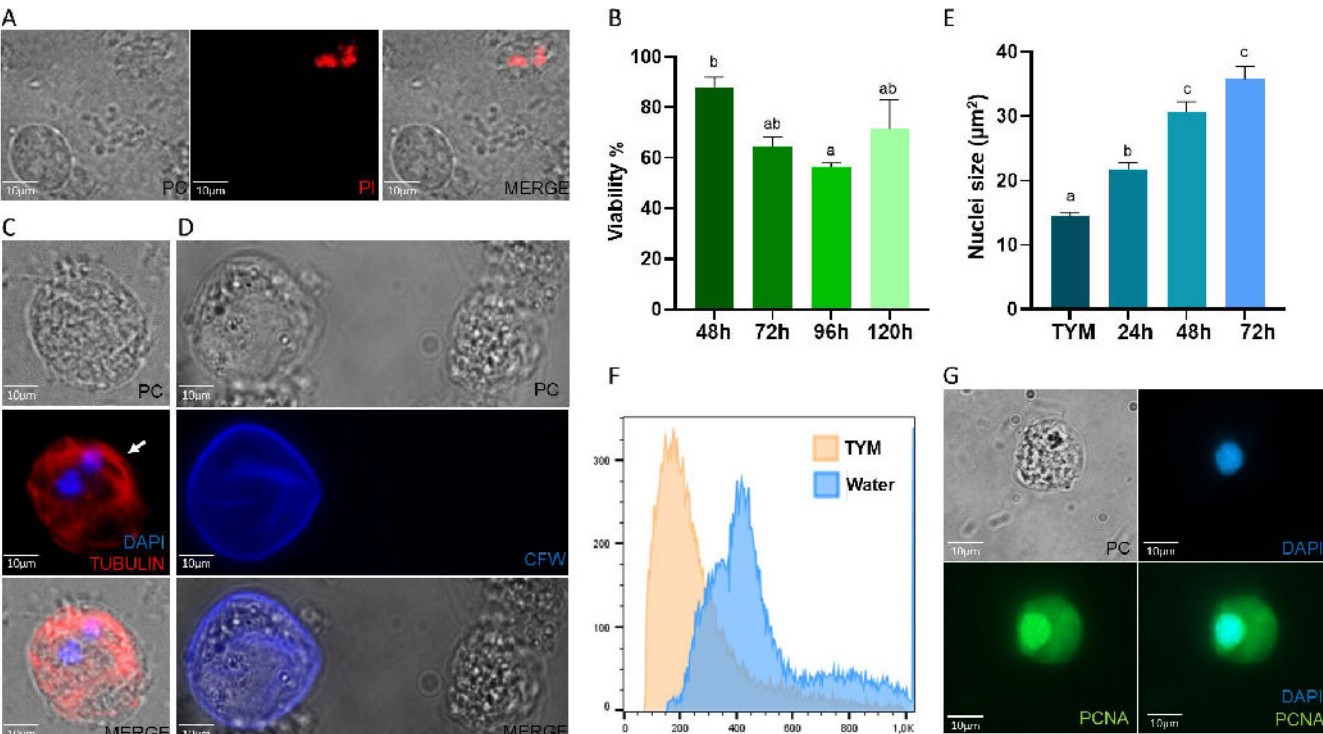

FIG 6 *T. foetus* water survival analysis. (A) A representative image of the viability of *T. foetus* maintained in water was determined by propidium iodide (PI) staining. Parasites were stained with PI and photographed under a fluorescent microscope. Alive parasites are not stained, whereas dead cells are stained red. (B) Quantification of the viability of *T. foetus* when they were incubated in water for 24, 48, 72, 96, and 120 h. One hundred parasites were counted in triplicate in three independent experiments. Bars represent the means of three independent assays. Samples were analyzed with ANOVA test. Mean differences were calculated using Tukey's test with an alpha = 0. 05. (C) Estimation of the size of the nuclei of *T. foetus* grown for 24, 48, and 72 h in water. TYM: parasites *T. foetus* grown in TYM culture media (control). The images of 50 nuclei staining with DAPI were analyzed using Image J's threshold tool and the "Analyze Particles" function. Experiments were performed in triplicate and repeated three times. Mean differences were calculated using the Kruskal-Wallis test with a *P* value of 0.05. (D) Indirect immunofluorescence using an anti-alpha tubulin antibody shows spherical parasites with internalized flagella (white arrow). DAPI (blue). PC, phase-contrast image (E) Representative images of a pseudocyst or cyst-like structure stained with Calcofluor white (CFW) in a *T. foetus* grown *in vitro* for 72 h, maintained in water. The samples were analyzed by epifluorescence microscopy. (F) A graph representing the DNA content profile of *T. foetus* grown in standard culture media conditions for 24 h (TYM) and *T. foetus* incubated in water for 24 h (Water). DNA content was measured by flow cytometry. (G) Image representative of an indirect immunofluorescence image showing PCNA-positive nuclei (green) of *T. foetus* parasites that were grown in water for 24 h. DAPI (blue). PC, phase-contrast image.

change could also be due to osmotic shock, we analyzed the cytosqueletal structures disposition by an indirect immunofluorescence assay using an antibody anti-tubulin, and we could observe that some of these spherical parasites were displaying typical pseudocyst features: internalized flagella and curved microtubular structures, such as the axostyle (Fig. 6C). Additionally, to further confirm whether these rounded parasites were pseudocysts or cyst-like structures, the parasites were fixed, stained with CFW, and analyzed by fluorescence microscopy. Parasites with surface CFW staining were observed (Fig. 6D), indicating the presence of pseudocysts or cyst-like structures in the water.

Considering that under certain stress conditions, the nuclei of *T. foetus* are capable of endoreplicating their DNA and that this process is directly related to an increase in nuclear size (27), we analyzed the size of the nuclei of *T. foetus* grown for 24, 48, and 72 h in water. The average nuclei size for parasites grown in standard culture media (control) was 14.46 μm², and for *T. foetus* parasites grown for 24, 48, and 72 h in water, it was 21.73, 30.71, and 35.79 μm², respectively (Fig. 6E). Next, the DNA content of parasites incubated in water (W) for 24 h was analyzed using propidium iodide (PI) staining by flow cytometry. As a control, the DNA content of *T. foetus* parasites grown for 24 h in standard culture media (TYM) was measured. As can be seen in Fig. 6F, the

parasites incubated in water contain more than two timesthe DNA content. Finally, we corroborated the active DNA replication in parasites incubated in water for 24 h by an indirect immunofluorescence assay using an antibody that specifically recognizes the "proliferating cell nuclear antigen" (anti-PCNA), which is a cofactor of DNA polymerases (27, 31, 32). We detected PCNA-positive *T. foetus* when parasites were grown under these conditions (Fig. 6G), indicating that the replication machinery is active in these parasites. These results were consistent with previous results where, under unfavorable conditions, the parasites increased their DNA content (33, 34). Our results demonstrated that *T. foetus* is able to survive several days in water, increase the DNA content and some of these parasites form pseudocysts or cyst-like structures.

## *T. foetus* is able to survive under pH conditions usually found in the bovine gastrointestinal tract

The main distinction in a bovine's digestive system is that the stomach has four separate compartments, each with a unique function. Reticulum-rumen, pH ranged between 5.5 and 7, is the place where fermentation occurs. The omasum absorbs nutrients and liquids, and the abomasum is used for acid digestion (pH 2–2.5). Most of the absorption of nutrients occurs in the duodenum and ileum, whereas there is also digestion and absorption of nutrients in the large intestine (pH 6–7). Considering that the whole digestion process takes about 48–72 h and the pH range of the digestive tract (Fig. 7A),

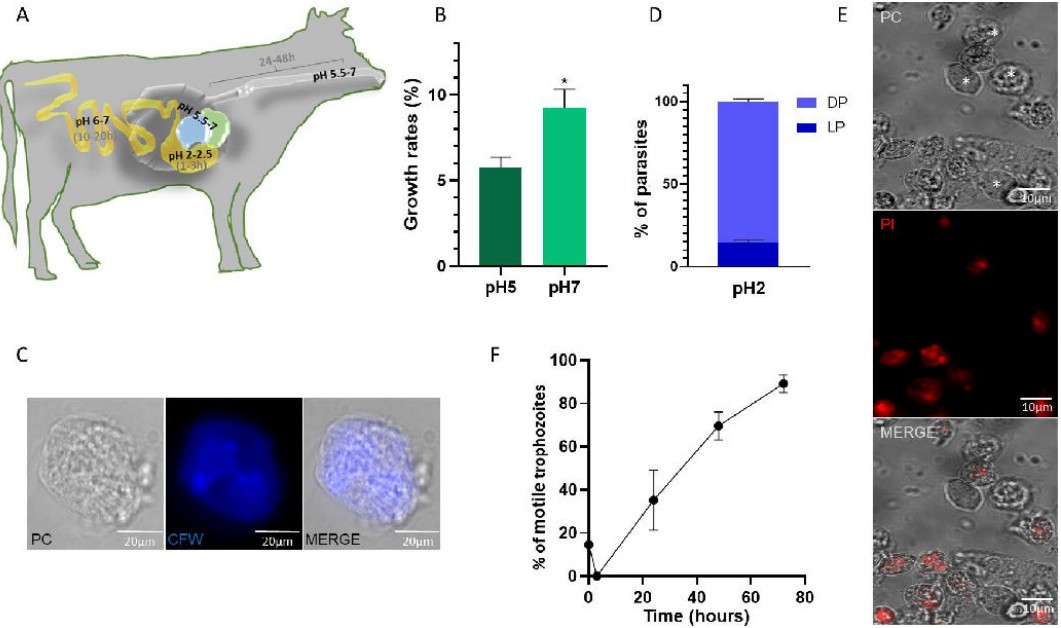

**FIG 7** *T. foetus* pH survival analysis. (A) Simplified schematic diagram of the ruminant digestive tract and pH conditions in various regions. The pH of the digestive tract is variable in the different regions: pH 5.5 and 7 (esophagus-reticulum-rumen), pH 2–2.5 (abomasum), and pH 6–7 (large intestine). The digestive transit times are as follows: 24–48 h for the esophagus, reticulum, and rumen; 1–3 h for the abomasum; and 10–20 h for the large intestine. (B) The growth rates of parasites grown in culture medium at pH 5 and 7 were compared. Error bars represent standard error, and asterisks denote statistically significant differences determined by *t*-Student ($P < 0.05$). (C) A representative image of a parasite stained with Calcofluor white (CFW) in a *T. foetus* grown in culture media at pH 2 for 3 h. The samples were analyzed by epifluorescence microscopy. (D) Quantification of viable *T. foetus* incubated for 3 h in TYM culture media at pH 2. One hundred parasites were counted in triplicate in three independent experiments. Bars represent the means of three independent assays. Samples show significant differences when analyzed with the Kruskal-Wallis test. LP (live parasites). DP (dead parasites). (E) A representative image of the viability of *T. foetus* maintained in water was determined by propidium iodide (PI) staining. Parasites were stained with PI and photographed under a fluorescent microscope. Alive parasites are not stained (white asterisks), whereas dead cells are stained red. (F) *T. foetus* kinetic growth curve depicting the percentage of motile trophozoites. *T. foetus* parasites were incubated in water for 24 h, then incubated for 3 h in culture media at pH 2, and then recovered in Diamond's medium for 24, 48, and 72 h. Parasite counts were collected at the indicated times on the *X*-axis. Each data point on the graph represents the average percentage of motile parasites from three independent experiments. The error bars indicate the standard deviation.

we examined if *T. foetus* could survive in the different pH conditions in standard TYM culture media. The results showed that the growth rate was greater in parasites grown in culture medium at pH 7 (9.23%), compared to parasites grown in culture medium at pH 5 (5.75%) (Fig. 7B). Taking into account that the inactivation of pathogenic microorganisms in mammals takes place in the stomach, an environment that is characterized by a low pH, we analyzed the pH 2 conditions in more detail. In this experiment, the parasites were pre-incubated in water for 24 h and then incubated at pH 2 condition, considering that if the parasites were ingested from the environment, they would be subjected to stress conditions prior to ingestion. While most of the parasites showed lysis when incubated at pH 2 for 3 h (time that takes for food to move through the omasum/abomasum), some parasites changed its form to a spherical shape with a diffuse CFW staining (Fig. 7C). Viability assays using propidium iodide staining demonstrated that 14.66% of parasites were viable when incubated 3 h at pH 2 (Fig. 7D and E). Finally, to evaluate if *T. foetus* parasites incubated for 24 h in water and then in TYM medium at pH 2 for 3 h could grow normally after treatment, they were transferred to TYM medium at pH 6.2 (the standard growth condition) and incubated for 24, 48, and 72 h (Fig. 7F). These results indicate that parasites maintained in stress conditions survive and are capable of multiplying when external optimal conditions are restored.

### *Tritrichomonas foetus* forms cyst-like structures

As we mentioned previously, the trophozoites of *T. foetus* (polar and flagellated) can adopt a spherical shape with internalized flagella, known as "pseudocystic" (as they do not possess a true cyst wall) under unfavorable environmental conditions, whose role in the life cycle of this protozoan is still unknown (2, 3, 35). In this context, it has been reported in some protozoans that there are different resistant dormant stages, namely cysts or pseudocysts (36, 37). Interestingly, the most evident differences between *Acanthamoeba* cysts and pseudocysts are the velocity of the cell response to stress and the structure and composition of the envelopes formed on the surface of these resting stages (a double-layered wall on the cyst or a single-layered fibrillar coat on the pseudocyst). The presence of chitin is a characteristic feature of cysts in other protozoan parasites like *Entamoeba invadens* and *Giardia lamblia* (38, 39) and Calcofluor White can be used to stain and detect cysts of certain parasites, particularly those that have chitin or cellulose in their cyst walls (27–29, 40). Here, we hypothesized that pseudocysts are mature cyst-like structures in formation in *T. foetus*. Previous reports about the evolution of cyst wall formation in *E. histolytica* used premixed calcofluor (CFW) and Evans blue (EB) dyes and proposed different steps in the encystment process: proliferating trophozoites (CFW−/EB+ population) and dormant cysts (CFW+/EB− population) (29).

We used this combined staining technique to analyze parasites maintained under nutritional stress conditions for 48 h. We observed the presence of both EB+/CFW− structures and CFW+/EB− structures (Fig. 8A), which indicates the existence of a cyst true wall formation process in some of these parasites. Which were acquiring a complete cyst wall structure (like mature cysts) and were losing permeability to solutes, such as EB. To confirm this result, taking into account that cyst-like forms are resistant to ionic detergents, *T. foetus* parasites grown under nutritional stress conditions for 48 h (Fig. 8B) were treated with 0.15% sarkosyl for 10 min (41). Sarkosyl-resistant parasites (cyst-like structures) were observed by microscopy (Fig. 8C). Our results demonstrate that *T. foetus* is capable of forming cyst-like structures under certain stress conditions.

## DISCUSSION

Trichomonads represent a group of protozoan flagellates adapted to live in anaerobic and microaerobic environments. Most of these organisms inhabit the intestines of a variety of vertebrate and invertebrate hosts, but only a few trichomonad organisms have been shown to have pathogenic potential for mammals and birds (42, 43). Nowadays, bovine tritrichomonosis is considered a venereal disease with important consequences for reproduction. In this context, the widespread use of artificial insemination in many

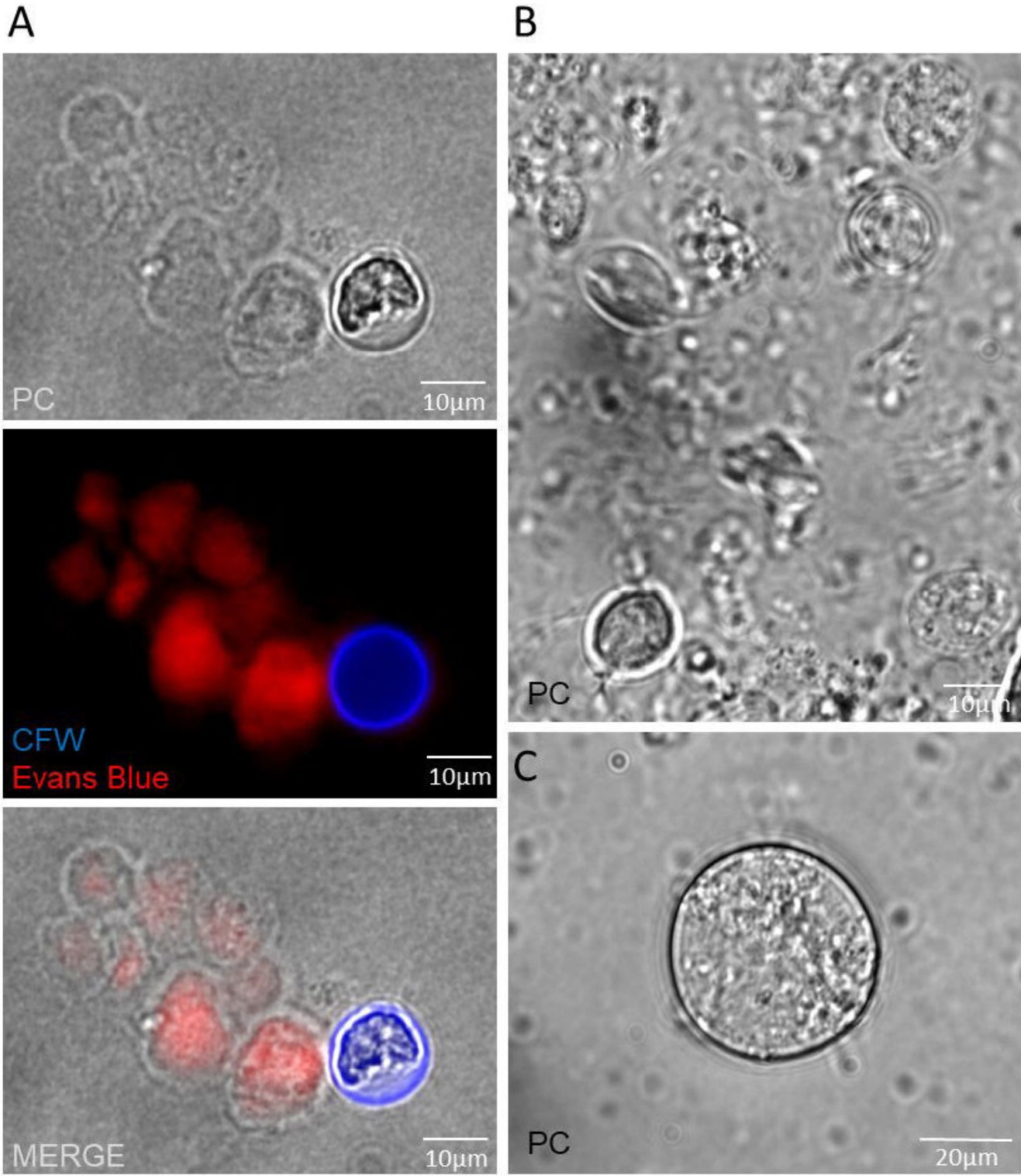

**FIG 8** *T. foetus* forms cyst-like structures. (A) A representative image of *T. foetus* parasites incubated in water (nutrient depletion) stained with a premixed reagent of Calcofluor White and Evans Blue dyes. Parasites were photographed under a fluorescent microscope. CFW (blue). Evans Blue (red). PC, phase-contrast image. (B) A representative image of *T. foetus* parasites incubated in water (nutrient depletion). PC, phase-contrast image. (C) Image of a cyst-like *T. foetus* resistant to ionic detergents, obtained after treating parasites previously incubated in water with sarkosyl. PC, phase-contrast image.

areas of the world has contributed to its reduced prevalence. Nevertheless, tritrichomonosis is still of importance in countries with extensive farming practices where artificial insemination is not used (4, 44). In extensively managed herds with natural service, bulls remain generally persistently infected and are considered the main reservoir for the parasite. Taking into account the current known life cycle of *T. foetus*, the usual diagnostic test is based on cell culture followed by microscopic examination to evaluate growth using samples collected from the prepuces of bulls. If motile trophozoites are observed, a PCR for confirmative diagnosis is performed. Hence, the control programs are worldwide focused on identifying and culling infected bulls and non-pregnant cows.

Despite the fact that bovine tritrichomonosis is thought to be only transmitted sexually, there are reports of trichomonads from bulls that have never mated in some regions of the world (11). In this context, it is interesting to consider that some Trichomonads such as *Tritrichomonas enteris*, *Tetratrichomonas buttreyi*, *Tetratrichomonas pavlova*, *Pentatrichomonas hominis*, and *Pseudotrichomonas* are usually found in the bovine gastrointestinal tract and often contaminate the preputial samples obtained to perform the *T. foetus* diagnostic tests, especially in those countries where extensive livestock production systems predominate (9, 45). The presence of these commensal trichomonad species in preputial samples frequently leads to misdiagnosis that could cause unnecessary culling of suspected *T. foetus*-infected animals. Curiously, previous reports postulated that gastrointestinal trichomonads are discharged by feces, reaching the prepuce of the other animal by sodomy, a common practice among young bulls (19). Afterward, these trichomonads could be transmitted from bulls to cows by coitus. Also, it has been suggested that the female reproductive tract could be auto-contaminated with gastrointestinal inhabitants contained in feces (20). Members of the trichomonads are also found in the avian digestive tract where they can cause a wide range of diseases, from subclinical to fatal infections (42). Besides, *Tritrichomonas muris* inhabits the gastrointestinal of mice, *Trichomonas equi* in horses, and *P. hominis* inhabits the gastrointestinal tract of a variety of vertebrate species (46, 47). Importantly, feline tritrichomonosis is a disease with a purported fecal-oral route of spread caused by *T. foetus* (48). Similarly, the presence of *T. foetus* in the gastrointestinal tract has also been reported in dogs with or without diarrhea (12). Further, *T. foetus* is a symbiont in the nasal passages, stomach, cecum, and colon of swine hosts (49). Also, this parasite can survive passage through the alimentary tract of slugs, and it has been proposed that slug feces could contaminate cat food and that this could be a route of infection for these animals (50), which demonstrates the great adaptability of this protozoan to different contexts in different hosts and mainly a special adaptation to the gastrointestinal tract. *T. foetus* has also been reported to cause infections in immunocompromised or immunosuppressed humans (51–54). And, it has been suggested that *T. foetus* may be capable of infecting patients orally as well as through direct contact with infected animals via *T. foetus*-contaminated natural raw foods (55).

*T. foetus* trophozoites have demonstrated a great capacity of survival in different environmental conditions. As a possible route for transmission, the parasite can survive more than 3 h in feline urine and sauced cat food, 2 h on ground cat food, and half an hour in tap or distilled water (56). In this context, the infection of turkeys and chickens happens mainly through ingestion of contaminated water by pigeons (46). Facilitating its transmission, different authors demonstrated the presence of *T. foetus* for several days at room temperature in feces with a normal or dry and firm consistency in cats (57). *T. gallinarum* in poultry species can be transmitted via consumption of contaminated food (46). Here, we showed that *T. foetus* is able to survive in an extract from bovine feces for several days at room temperature and for 120 h in water, takes on a pseudocyst or cyst-like form, as well as in the different pH conditions usually found throughout the bovine digestive tract. In this sense, the trichomonads possess hydrogenosomes that produce molecular hydrogen in anaerobiosis and reduce oxygen, whereby the parasite could manage better pH variations favoring its own development (58). In this regard, it has been shown that the encystation of *Trichomonas vaginalis* trophozoites occurs

under acidic pH conditions, a similar condition encountered by the parasite in the human vagina. The authors propose that the low vaginal pH might trigger the parasite's transformation from trophozoites to cyst-like structures (28). Related to this, it is known that *T. foetus* is capable of surviving the acidic pH levels in the stomachs of pigs and cats (12, 14, 22) and it has been reported the existence of *T. foetus* pseudocysts in feline feces (59).

Considering that the moist rumen environment requires large quantities of water and bovines drink up to 100 liters a day; the ingestion of water contaminated with feces containing parasites could be a route of infection that has never been considered for *T. foetus* transmission. Interestingly, it has also been reported that *T. vaginalis* survives in swimming pool water for several hours (60), and that *T. foetus* is capable of surviving at least 24 h at 4°C (61). Finally, it has been shown that *T. vaginalis* cyst-like structures are resistant to osmotic variation in their environment (28). Trophozoites and pseudocysts are the two cellular forms known for *T. foetus*. It has been described that pseudocysts are multinucleated forms induced by stress situations such as temperature variation, iron depletion, and chemical treatment (35, 61–63). In this context, it has been demonstrated that in preputial secretions from *T. foetus*-infected bulls, the presence of the pseudocyst form occurs more frequently than the pear-shaped form (62), and such pseudocysts are more cytotoxic when in contact with host cells (64). Remarkably, the cyst-like stages (pseudocysts) of *T. gallinae* (65) have been proposed as a resistant form of parasite survival during fecal-oral transmission (66). Additionally, it has been shown that *T. muris* fecal-oral transmission occurs via ingestion of either the pseudocyst or trophozoite stage of this parasite (67). Related to this, we observed that *T. foetus* is capable of surviving in the gastrointestinal tract of bovines and being discharged viable in feces. Moreover, we demonstrated the presence of structures stained with Calcofluor White Stain (fluorochrome that binds to the pseudocysts/cyst-like structures surface) in parasites incubated both in extract from feces and maintained in water. These pseudocysts/cyst-like structures may represent an environmentally resistant stage during unfavorable conditions that could provide a novel route of parasite transmission.

It is known that during environmental stress, the vegetative cells of the facultative pathogenic amoeba *A. castellanii* reversibly differentiate into resistant dormant stages (cysts or pseudocysts), depending on the nature and duration of the stressor. In common, both of these resting stages are able to convert back to the proliferating state shortly after the external stress stimulus passes away. The most evident differences between Acanthamoeba cysts and pseudocysts are the velocity of the cell response to stress and the structure and composition of the envelopes formed on the surface of these resting stages (a double-layered wall on the cyst or a single-layered fibrillar coat on the pseudocyst) (37). In *E. histolytica*, the main differentiation processes of encystation involve proliferating trophozoites and immature or mature cysts, depending on the cyst's complete wall presence (29). In *Histomonas meleagridis*, precystic stages, pseudocysts, and true cysts have been described that survive in adverse conditions, including extreme acid conditions (pH 1–2 for approximately 4 h) (68, 69). Transformation of typical trophozoites to pseudocysts has also been observed in *Trichomonas vaginalis* (2, 70, 71), *Tritrichomonas muris*, rodent gut colonizers (72), and *Trichomonas tenax*, a commensal organism (73).

In the current study, we have characterized the cyst-like structure using two fluorescent dyes that uniquely bind to chitin: Calcofluor White (74) and Alexafluor-conjugated Wheat Germ Agglutinin, or WGA (75). We have demonstrated that *T. foetus* cyst-like structures, in contrast to trophozoites, were resistant to detergent and did not allow the diffusion of Evans blue into the interior of the cell. The observation agrees well with a previous study that identified chitin on the cell surface of *T. vaginalis* and *T. foetus* using anti-chitin antibodies (76).

We thus propose the nomenclature of these forms as cyst-like structures rather than pseudocysts, as these forms have the hallmarks of true cysts. Moreover, we hypothesized that pseudocysts observed in *T. foetus* (with diffuse CFW stain) could be cysts with

incomplete chitin walls, which represent the intermediate stages in chitin wall formation as they occur in *Entamoeba* and *Acanthamoeba* cyst wall formation (75, 77). Although, it is necessary to evaluate the formation process and the role of these structures in *T. foetus*' transmission in the future.

In summary, we demonstrated that *T. foetus* is more resilient than previously believed, and this may have important implications for the epidemiology of bovine tritrichomonosis. In this context, Iriarte et al. (27) demonstrated that *T. foetus* under stress conditions is capable of increasing its DNA content per parasite without concluding the cytokinesis process (endoreplication), which represents an efficient strategy for subsequent fast multiplication by multiple fission when the context becomes favorable. Additionally, they revealed the existence of novel dormant forms of resistance (multinucleated or mononucleated polyploid parasites), different than the previously described pseudocysts, that are formed under stress conditions.

Considering that *T. foetus* is capable of surviving in the bovine gastrointestinal tract conditions, that the parasite could be discharged by feces into the environment, and their capacity to form cyst-like structures (resistance structures), we propose a novel possible dissemination form of *T. foetus*, mainly in extensive livestock production systems of bovines, e.g., in Argentina or the USA (4, 44, 78) (Fig. 9), which should be analyzed in detail in the future.

Grazing and water puddles contaminated with feces containing parasites could be ingested by cows and bulls (Fig. 9, number 1) and the parasite discharged by feces

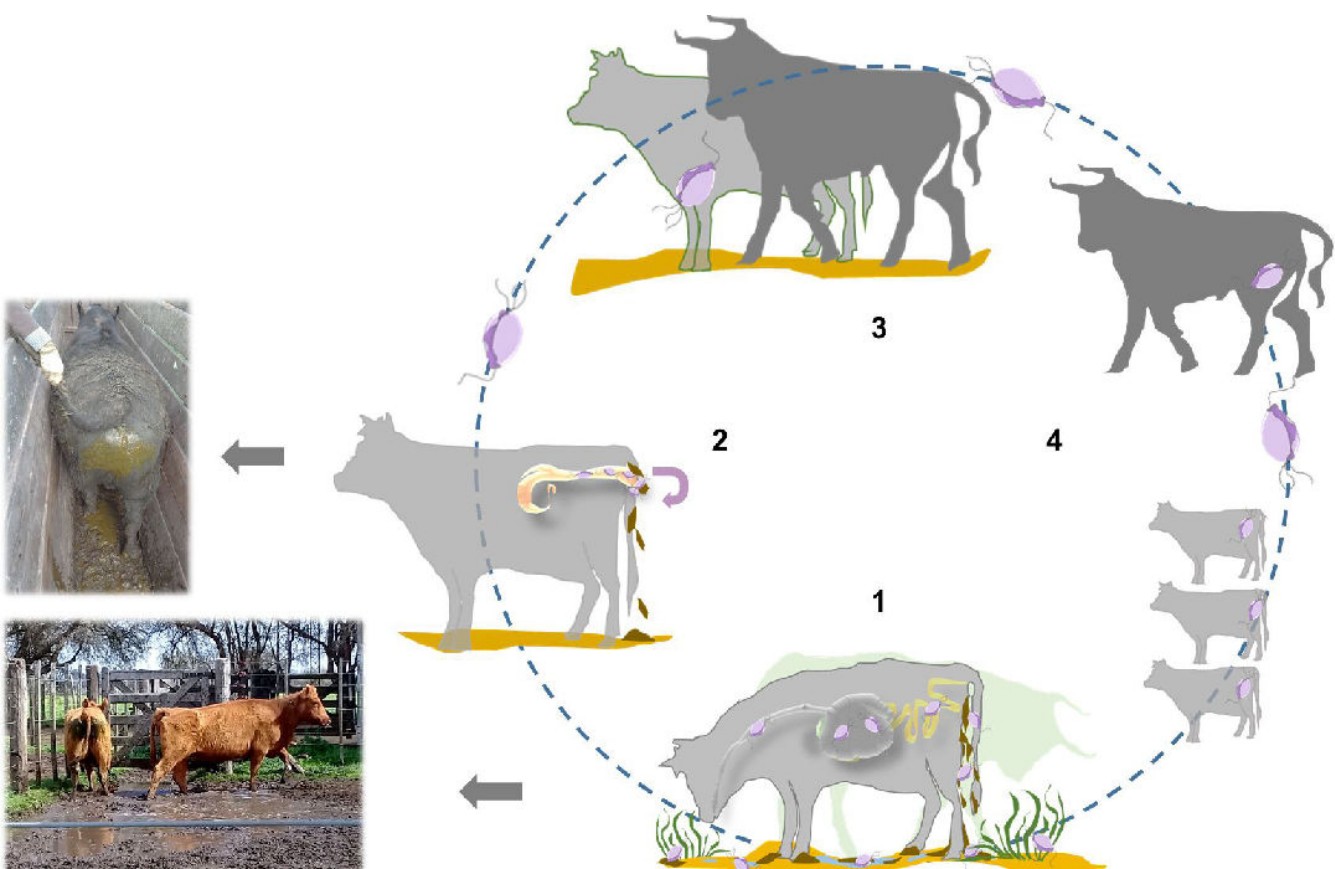

**FIG 9** A novel possible route for dissemination of the *T. foetus* in bovines. *T. foetus* is capable of surviving in the bovine gastrointestinal tract and could be discharged by feces into the environment. Grazing and water puddles contaminated with feces containing parasites could be ingested by cows and bulls (1) and the parasite discharged by feces (2). Contamination of the female reproductive tract with feces-containing parasites could occur during the breeding season (2). In cows, self-contamination of the female reproductive tract with feces containing the parasite could occur (2). Alternatively, *T. foetus* present in feces could reach the cow's vagina and/or bull's preputial mucosa during the coitus (3). Infected bulls can disseminate the infection by transmission to other cows during the breeding season (4).

(Fig. 9, number 2). As breeding season coincides with the new greener pastures with higher moisture content, the animals' feces become soft and even liquid. In cows, self-contamination of the female reproductive tract with feces containing the parasite could occur due to the vulva's ventral position with respect to the anus in the perineal region (Fig. 9, number 2). Alternatively, *T. foetus* present in feces could reach the cow's vagina and /or bull's preputial mucosa during the coitus (Fig. 9, number 3). As regularly as each bull serves 25–30 cows in an extensive production system, infected bulls can disseminate the infection by transmission to other cows during the breeding season (Fig. 9, number 4).

While venereal transmission of bovine trichomonosis is the most significant form of infection, our results require rethinking the current management and control of this parasitic disease in extensive livestock production systems. Nowadays, prevention and control of *T. foetus* transmission are based on identifying infected animals and culling practices. The routine herd diagnosis is usually performed exclusively on bulls, as they remain permanently infected. However, the results obtained here highlight the need to reformulate and evaluate alternative programs to control bovine tritrichomosis in the future in these livestock production systems.

## MATERIALS AND METHODS

### Parasites cultures

*T. foetus* K strain (Embrapa, Rio de Janeiro, Brazil) was cultured in Diamond's TYM medium supplemented with 10% horse serum and 10 U/mL penicillin/10 µg/mL streptomycin (Invitrogen, Buenos Aires, Argentina) at pH 6.2. Parasites were grown at 37°C and passaged daily. 1N NaOH and 5N HCL were used to adjust the pH of the media to the desired value.

### Nuclear staining

The number of nuclei per cell was determined using DAPI (4′, 6-diamidino-2-phenylin-dole) staining. Parasites in the absence of host cells were incubated (in TYM culture media) at 37°C on glass coverslips for 4 h, then were fixed and permeabilized in cold methanol for 10 min. The cells were washed three times in PBS and incubated with a 300 nM DAPI stain solution for 5 min, protected from light. After three washes with PBS, the coverslips were mounted onto microscope slides using fluoromont mounting media (Sigma-Aldrich, Buenos Aires, Argentina). Experiments were performed in triplicate and repeated three times.

Images were acquired using fluorescence microscope (Zeiss Axio Observer 7 inverted fluorescence microscope, USA) and the nuclear area in µm$^2$ was examined using ImageJ's threshold tool and the "Analyze Particles" function, and the experiments were carried out in triplicate and three times.

### Wheat germ agglutinin binding assay

Parasites in TYM culture media were incubated at 37°C on glass coverslips for 1 h and then fixed in 4% paraformaldehyde at room temperature for 20 min. Then, parasites were incubated at 37°C in a 1:100 of Lectin from *Triticum vulgaris* (WGA, Sigma-Aldrich) conjugated with FITC/phosphate-buffered saline dilution for 1 h, and then the parasites were washed three times with PBS. The slides were then washed three times in PBS, mounted on fluoromont (Sigma-Aldrich), and observed under a phase contrast and fluorescence microscope (Zeiss Axio Observer 7 inverted fluorescence microscope).

## Calcofluor white staining

Parasites were pipetted onto a glass slide, air-dried, and fixed in methanol for 10 min at room temperature. Slides were washed three times in PBS for 5 min each time and incubated with 0.01% Calcofluor White stain (comprising of 1 g/L CFW M2R and 0.5 g/L Evans blue) (Sigma-Aldrich) in PBS, pH 7.2, for 30 min at 26°C. The slides were then washed three times in PBS, mounted on fluoromont (Sigma-Aldrich), and observed under a phase contrast and fluorescence microscope (Zeiss Axio Observer 7 inverted fluorescence microscope).

## Detergent resistance assay

For the detergent resistance assay, $1 \times 10^6$ trophozoites pre-incubated in water for 48 h were centrifuged for 10 min at room temperature, and total cells were treated with 0.15% sarkosyl for 10 min. Then, cyst-like structures resistant to detergents were washed three times with PBS and observed by microscopy (Zeiss Axio Observer 7 inverted fluorescence microscope).

## Polymerase chain reaction

To analyze the protozoans isolated from feces and mucus samples, the samples were previously cultured in Diamond's TYM medium supplemented with 10% horse serum and 10 U/mL penicillin/10 µg/mL streptomycin (Invitrogen) at pH 6.2. The protozoans were recorded every 24 h using a Neubauer hemocytometer. When the protozoans reached the exponential growth phase, $1 \times 10^5$ protozoans were centrifuged at 5000 $\times$ $g$ for 5 min at 4°C. The pellets were rinsed three times with ice-cold physiologic solution (0.85% NaCl) and resuspended in 100 µL of physiologic solution. A PCR assay was performed using the primers TFR-3 (5′-CGGGTCTTCCTATATGAGACAGAACC-3′) and TFR-4 (5′-CCTGCCGTTGGATCAGTTTCGTTAA3′). The PCR reactions were performed with 1× Taq buffer (Invitrogen), 2.5 mM MgCl$_2$, 0.2 mM of each dNTPs, 1 µM of specific primers, 1 µL of the sample, and 1.25 U of Taq DNA polymerase (Invitrogen) in a final reaction volume of 50 µL. The assay was run through 30 cycles of denaturing (94°C, 30 s), annealing (67°C, 30 s), and extension (72°C, 60 s). Amplification was performed on an Applied Biosystems Thermal Cycler, USA. The amplified products and base pair size markers were electrophoresed on 1% agarose gels, stained with ethidium bromide, and photographed using an ultraviolet light transilluminator.

## Sequence analysis

Multiple sequence alignment (MSA) was performed by MUSCLE algorithm (v3.8.31) using default parameters (79). JalView viewer was implemented for visual and aesthetics, and the percent identity matrix calculation (80).

## Cell viability assay

*T. foetus* parasites were centrifuged and resuspended in 40 µg/mL propidium iodide solution (a membrane impermeant dye that is excluded from viable cells), for 30 min at 37°C. Percentage of viable cells was analyzed using an epifluorescence microscope (Zeiss Axio Observer 7 inverted fluorescence microscope). Experiments were performed in triplicate and repeated three times.

## Flow cytometry

To determine DNA content, parasites ($5 \times 10^6$) were harvested and fixed in 5 mL of ice-cold 100% EtOH at 4°C overnight. Following that, each sample was washed in 1 mL of PBS containing 2% vol/vol horse serum (HS), resuspended in 1 mL of PBS containing 180 µg/mL RNase A (Sigma-Aldrich) to digest RNA and 2% vol/vol HS, and incubated for 30 min at 37°C. Then, samples were stained with a 25 µg/mL PI solution and incubated for 30 min at 37°C prior to flow cytometer analysis. Flow cytometry analysis was carried

out using a FACS Calibur flow cytometer (Becton Dickinson, San Jose, USA) equipped with a dual laser system (15 mW 488 nm argon ion laser and a 635 nm red diode laser). For measurement of DNA content, cells were excited with 480 nm light and emission was measured through 585/42 (for PI fluorescence; FL2). Data from 20,000 cells were recorded, and these sets were analyzed using the FlowJo 7.6 software. Correlation between the light scattering properties of cells with their DNA content was performed by setting electronic gates on the forward scatter (FSC): side scatter (SSC) profiles and checking for DNA content of every gate.

### *Tritrichomonas foetus* gastrointestinal transit evaluation

To evaluate the possible elimination of live *T. foetus* by feces, $1 \times 10^8$ parasites resuspended in PBS buffer were administered orally for 5 days to two cows (*T. foetus* negative). After the third day of initial oral administration, the feces samples were collected per rectum during 7 consecutive days. These 14 samples incubated in separate tubes and per triplicate were homogenized by mixing them in TYM media supplemented with 10% fetal horse serum + penicillin/streptomycin + amphotericin B (5 g feces in 5 mL media) and incubated at 37°C for up to 12 days. The presence of trophozoites of *T. foetus* (approx. 20 µm length × 10 µm width; fusiform shape) was evaluated daily by direct microscopic examination. As a control, feces samples were obtained before the treatment.

### Sampling and analysis of animals

Preputial and vaginal mucus (from five bull and five non-pregnant cows) scraping samples were collected of bovines from a cattle farm where bulls has been diagnosed positive for *T. foetus*. A total of 10 fecal samples (5 bulls and five cows) were collected per rectum. Then, the samples were cultivated per triplicate in TYM media supplemented with 10% fetal horse serum. All cultures were incubated at 37°C and checked daily for up to 15 days for the presence of motile flagellated protozoans.

### Assay for parasite growth in an extract from bovine feces

To analyze whether *T. foetus* was capable of surviving in bovine feces, feces from 10 bovines were thoroughly homogenized by mixing in PBS (5 g of feces/5 mL of PBS). Then, the mixture was centrifuged and the supernatants, or "extract from bovine feces," were obtained. Afterward, $1 \times 10^5$ parasites were centrifuged and resuspended in 100 µL of PBS (*T. foetus K* strain), and afterward were inoculated in 15 mL of the supernatants obtained from the mixture (feces/PBS) and incubated at room temperature. After inoculation, cell counts were recorded every 24 h using a Neubauer hemocytometer and the kinetics of growth curves were performed.

The growth rates were determined as the natural logarithm of the change in the density of parasites per milliliter at time *t* compared with that at time 0 (initial inoculum) by the following equation: growth rate = [lnCC (*t*) − lnCC (0)]/ (t − 0), where CC (*t*) and CC (0) are the parasites counts per milliliter at time *t* and time 0, respectively, and *t* is the time of incubation (81).

### Analysis of *T. foetus* survival in water

A total of $1 \times 10^6$ *T. foetus* parasites were kept at room temperature for 24, 48, 72, and 120 h in water and 10 U/mL penicillin/10 g/mL streptomycin (Invitrogen). The parasites were centrifuged, resuspended in a 40 ug/mL propidium iodide solution, incubated for 30 min at 37°C and counted using an epifluorescence microscope (Zeiss Axio Observer 7 inverted fluorescence microscope). Experiments were performed in triplicate and repeated three times.

## Analysis of *T. foetus* survival in different pH conditions

*T. foetus* K strain at a concentration of $1 \times 10^6$ was cultured in TYM medium supplemented with 10% fetal horse serum and 10 U/mL penicillin/10 µg/mL streptomycin (Invitrogen) at pH 5 and 7. The parasites were grown at 37°C and passaged daily, and their growth was monitored until their death phase. Besides, the *T. foetus* K strain pre-incubated in water for 24 h was cultured in TYM media at pH 2 for 3 h at 37°C to perform viability assays using the propidium iodide protocol.

## Immunolocalization experiments

Parasites, grown in different conditions, were incubated at 37°C on glass coverslips for 4 h and then fixed and permeabilized in cold methanol for 10 min. Cells were then washed and blocked with 5% fetal bovine serum (FBS) in PBS for 30 min, incubated with a 1:500 dilution of anti-alpha tubulin primary antibody (Sigma-Aldrich) or anti-PCNA monoclonal antibody (Abcam, Tecnolab S.A, Buenos Aires, Argentina) (1:400) diluted in PBS plus 2% FBS for 2 h at RT, washed with PBS, and then incubated with a 1:5,000 dilution of Alexa Fluor-conjugated secondary antibody (Molecular Probes) 1 h at RT. The coverslips were mounted onto microscope slips using ProLong Gold antifade reagent with 4, 6′-diamidino-2- phenylindole (Invitrogen). All observations were performed on an epifluorescence microscope (Zeiss Axio Observer 7 inverted fluorescence microscope). Adobe Photoshop (Adobe Systems) was used for image processing.

## Scanning electron microscopy

Cells were washed with PBS and fixed in 2.5% glutaraldehyde in 0.1 M cacodylate buffer, pH 7.2. The cells were then post-fixed for 15 min in 1% $OsO_4$, dehydrated in ethanol and critical point dried with liquid $CO_2$. The dried cells were coated with gold–palladium to a thickness of 15 nm and then observed with a Jeol JSM-5600 scanning electron microscope, operating at 15 kV.

### AUTHOR AFFILIATIONS

[1]Laboratorio de Parásitos Anaerobios, Instituto Tecnológico Chascomús (INTECH), CONICET-UNSAM, Chascomús, Argentina
[2]Escuela de Bio y Nanotecnologías, Universidad Nacional de San Martin (UNSAM), Buenos Aires, Argentina
[3]Laboratorio de Parasitología Molecular, Instituto Tecnológico Chascomús (INTECH), CONICET-UNSAM, Chascomús, Argentina
[4]Centro de Diagnóstico e Investigaciones Veterinarias, FCV-UNLP, Chascomús, Argentina
[5]Departamento de Microbiologia, Fiocruz, Instituto Aggeu Magalhães, Recife, Pernambuco, Brazil

### AUTHOR ORCIDs

Natalia de Miguel  https://orcid.org/0000-0002-3864-0703
Veronica M. Coceres  http://orcid.org/0000-0002-2815-7953

### AUTHOR CONTRIBUTIONS

Cristian I. Martínez, Conceptualization, Formal analysis, Investigation, Methodology, Writing – original draft | Lucrecia S. Iriarte, Conceptualization, Data curation, Investigation | Nehuen Salas, Formal analysis, Writing – review and editing | Andrés M. Alonso, Formal analysis, Methodology | Cesar I. Pruzzo, Formal analysis, Writing – review and editing | Tuanne dos Santos Melo, Data curation, Writing – review and editing | Antonio Pereira-Neves, Conceptualization, Investigation, Writing – review and editing | Natalia de Miguel, Conceptualization, Formal analysis, Investigation, Writing – review and editing.

## ADDITIONAL FILES

The following material is available online.

### Open Peer Review

**PEER REVIEW HISTORY (review-history.pdf).** An accounting of the reviewer comments and feedback.

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
