## [Reviewer comments · Microbiology Spectrum]

Microbiology Spectrum

Prolonged survival of venereal *Tritrichomonas foetus* parasite in the gastrointestinal tract, bovine fecal extract, and water

Veronica Coceres, Cristian Martínez, Lucrecia Iriarte, Nehuen Salas, Andres Alonso, Cesar Pruzzo, Tuanne dos Santos Melo, Antonio Pereira-Neves, and Natalia de Miguel

Corresponding Author(s): Veronica Coceres, Instituto Tecnológico de Chascomus

Review Timeline:

Submission Date:	January 30, 2023
Editorial Decision:	April 20, 2023
Revision Received:	June 30, 2023
Editorial Decision:	August 7, 2023
Revision Received:	August 13, 2023
Editorial Decision:	August 15, 2023
Revision Received:	August 15, 2023
Accepted:	August 16, 2023

Editor: Neil Mabbott

Reviewer(s): The reviewers have opted to remain anonymous.

Transaction Report:

DOI: <https://doi.org/10.1128/spectrum.00429-23>

April 20, 2023

Dr. Veronica Mabel Coceres
Instituto Tecnológico de Chascomus
Laboratorio de Parásitos Anaerobios
Intendente Marino km 8.2
Chascomús, Buenos Aires 7130
Argentina

Re: Spectrum00429-23 (*Tritrichomonas foetus* life cycle: report of a new spread route in bovines)

Dear Dr. Veronica Mabel Coceres:

Thank you for submitting your manuscript to Microbiology Spectrum.

Your paper has been considered by three Reviewers and their comments are included below. You will see that each of the Reviewers has raised a series of important limitations that should be addressed in your revised manuscript. Based on these comments we would be willing to consider a thoroughly revised manuscript that addressed these concerns.

Link Not Available

Sincerely,

Prof Neil Mabbott

Journals Department
Reviewer comments:

Reviewer #1 (Comments for the Author):

In this manuscript the authors investigate the potential role of the gut as a site of infection for *Tritrichomonas foetus* in bovine. The author combine experimental infections experiments with survey of naturally infected bovine to demonstrate the gut-UGT infections can take place in addition to STI route of infection. This is a very interesting and important paper that resolves an important conundrum of some infection pattern among bovine that could not be simply explained by STI transmissions. It is also complementing existing data for oral-fecal route transmission of the same parasite in other hosts (cats, dogs, pigs). I have only a

few points to improve this overall very good manuscript.

Main points

It would have been important to sequence the PCR products detected from natural infections and compare the sequences with published sequences. As there is plenty of PCR products, and if they are still available, this should be straightforward to generate. How heterogeneous are these sequences and how do they compare with those from various isolates (e.g. cats, dogs, pigs), see for example Kleina et al. (2004) and related papers.

Kleina et al. (2004). Molecular phylogeny of Trichomonadidae family inferred from ITS-1, 5.8S rRNA and ITS-2 sequences. *International Journal for Parasitology*, 34: 963-970.

The sentence "Unlike other parasitic protozoa, *T. foetus* has only one morphological stage, the trophozoite." Is not accurate. Indeed, it is contradicted in the following sentence that is referring to the pseudocyst form. It would be more accurate, and more straightforward, to simply refer to trophozoite and pseudocysts as the two cellular forms known for that parasite. The related sentence in the discussion needs to be edited accordingly (section starting at line 320).

Similarly, the sentence "Thus, these organisms are frequently found in the samples obtained from bulls when analyzed by the standard diagnostic methods." Should be made more precise as the frequency of such gut derived infections are context dependent and have only rarely been detected in some contexts, such as in Europe. See for instance Bailey et al. (2021). Bailey et al., (2021). Sporadic isolation of *Tetratrichomonas* species from the cattle urogenital tract. *Parasitology*, 146: 1109-1115.

Note that in this paper that no *T. foetus* was detected in the gut metatranscriptomics data from another study.

When discussing the pseudocyst it is important to consider the observations by Pereira-Neves et al. (2012) that the pseudocyst were even more cytotoxic than the trophozoites. This could be relevant in the transmission from gut to the UGT, a transmission route suggested in this study.

Pereira-Neves A, Nascimento LF, Benchimol M (2012) Cyto-toxic effects exerted by *Tritrichomonas foetus* pseudocysts. *Protist*, 163:529-543.

Other points

T. foetus was also isolated from a few human patients, see review by Martiz et al. (2014) and more recently see paper by Zuzuki et al. (2016).

Martiz et al. (2014). What is the importance of zoonotic Trichomonads for human health? *Trends in Parasitology*, 30: 333-341.

Zuzuki et al. (2016). Characterization of a human isolate of *Tritrichomonas foetus* (cattle/swine genotype) infected by a zoonotic opportunistic infection. *Journal of Veterinary Medical Science* 78: 633-640.

pH3 conditions -> pH 3 conditions

Reviewer #2 (Comments for the Author):

This manuscript authored by Martinez CI and colleagues is addressing possibility that *Tritrichomonas foetus*, a trichomonad protozoa can be transmitted by fecal contamination of the urogenital tract of female cattle followed by sexual transmission. They had logically showed: first, that the trichomonad parasite survived and passed through the gastrointestinal tract, and was able to reach the urogenital tract of cows by fecal contamination following oral inoculation of trophozoites; second, the trophozoites of *T. foetus* was found in the fecal materials of naturally infected cattle; third, these trophozoites survived days in feces; fourth; they remained viable in water for up to 72 hours; last, but not the least, they tolerated low pH conditions mimicking the gastrointestinal tract of cattle. All these data were solidly laid out. Based on these data, they proposed that trophozoites that originate in the gastrointestinal tract and contaminate the urogenital tract of cows serve as a new spread route for the parasites to be sexually transmitted among cattle herds.

Whether this newly proposed route might be true needs to be confirmed. Nevertheless, some already well-known facts seem contradicting, or at least not supporting such a notion. First, the methods for prevention and control of bovine trichomonosis solely targeting the sexual transmission route have achieved eradication of the disease on state or even country wide scale.

Many European countries have achieved it by widespread use of artificial insemination (AI)

(<https://www.microbiologyresearch.org/content/journal/jmm/10.1099/jmm.0.047365-0#tab2>). Wyoming State of the United State America has reached eradication by testing bulls and culling *T. foetus*-positive bull

(<https://veterinaryresearch.biomedcentral.com/articles/10.1186/s13567-021-00996-w>). These successful achievements

undoubtedly pinpoint the sexual contact as the main, if not the only transmission route of the disease. Second, five cows were tested for *T. foetus* trophozoites in both the gastrointestinal and urogenital tracts. One animal was tested positive in both tracts, whereas the other two were only positive in the latter, which would be hard to be explained by fecal contamination. They seemed to be infected by sexual activities. To confirm the newly proposed route of transmission, an isolated herd of virgin heifer with infection of the parasites in the gastrointestinal tract is needed. These females are allowed to mate with *T. foetus*-negative bulls, and female heifers become positive after mating in the absence of the parasite in their feces. This reviewer understands this is

very challenging to achieve. Without such confirmative data your proposed notion of new transmission route is hard to believe due to the reason just mentioned. Even this route is proven to be true, it may just play a very minor role in some herds or no role at all in others. Therefore, it is of no great effect on the current methods for control and prevention.

You only mentioned the direct economic losses due to abortion and infertility. However, there are indirect losses as well such as testing fees and veterinarian costs. See the paper - <https://veterinaryresearch.biomedcentral.com/articles/10.1186/s13567-021-00996-w>

"T. foetus is also a commensal parasite found in the intestines and stomachs of pigs." This statement is not correct. A commensal CAN'T be a parasite, or vice versa by definition. They belong to very different categories.

A scale bar is needed for all micrographs in figures.

Figure 2C: It is not clear whether the samples were from same cow or different cows.

Figure 4A: In M&M: growth rate = $[\ln CC(t) - \ln CC(0)] / (t - 0)$, where CC (t) and CC (0) are the parasites counts per milliliter at time t and time zero, respectively, and t is the time of incubation (45)." This figure shows parasites/ml. Please explain the discrepancy. Figure 4E: Labels "TYM and BFE: are missing in the panels.

For equipment and reagents, please list providers and their location including city & country.

You described two ways to apply T. foetus to a cover slip for downstream application such as staining, one is direct; another is culturing trophozoites for hours. In culture, no medium was specified. Why did the trophozoites need to be cultured on a coverslip prior to staining etc.?

How were the trophozoites in feces and mucus samples quantified as mentioned in L409-410?

PCR primer concentration should not be 1 mM. It is a typo of 1 μ M, isn't it?

Only two cows were used in the experiments of gastrointestinal infection. The sample size is too small. How was the infection performed? What treatment was used to clear the infections?

References: the names of genus and species should be italicized. Also, the font size and line space should be the same as those of the main text.

L265: "microscopic examination followed by cell culture" The order seems reversed. Usually cell culture takes place first, followed by microscopy.

L267-268: AI should be mentioned as well since it is the most effective way to control and eradication of bovine trichomonosis, which has been proven in most European countries.

L467-468: In what a volume were 1×10^5 parasites (T. foetus K strain) inoculated in the supernatants?

Minor issues:

Be consistent, both tritrichomonosis and trichomonosis are used for the same disease caused by T. foetus.

The word "eliminate" seems not a right word to describe trophozoites in host feces, discharge or pass or secrete?

L17: "the life cycle of the protozoan T. foetus"

L43-44: "Unlike other parasitic protozoa, T. foetus has only one morphological stage, the trophozoite". The statement is inaccurate since quite a few species of protozoa have only trophozoite stage.

"T. foetus in samples from virgin bulls or the presence of bull negative samples and cow positive samples in the same bovine herds are clearly situations difficult to explain based on the current knowledge of T. foetus life cycle." The exact same sentences appear multiple times in the manuscript. Please reword them differently.

L76: "amphibian intestinal parasite,"

L77: "mammal intestine"

L273: "some trichomonads such"

L326: "T. muris fecal-oral transmission"

L345: "multiplication by multiple fission"

L572: "an alpha = 0.05."

Reviewer #3 (Comments for the Author):

This paper examines transit of T. foetus through the GI tract. The authors propose they have discovered a new mechanism whereby cows ingest oocysts, defecate them onto the vulva, and become infected AND possibly perpetuate the infection via sodomy.

The concatenation of experiments is somewhat confusing- in that the majority of the experiments do not show the lifecycle resulting in early embryonic death. Moreover, there are no experiments that show sexual transmission following ingestion. I think the paper is worthy of publication but it needs a major revision so that it is not over reaching in its conclusions. That is, what this paper really attempts to show, is transit of T. foetus through the bovine GI tract. What they do show, is that the parasite can survive in feces and at different pH. They show that in infected herds you can find PCR positive feces, and they show that if you feed cows excessive numbers of trich you can find a few in feces (1 of 5 cows and 1 of 5 bulls). And this is not surprising since the authors begin by talking about how T. foetus can transition through the GI system of pigs and of course cats.

Key references: those of infections where there is no sexual reproduction (no bull), etc, are completely lacking. Indeed the only references of connumdrums occur when a fecal flagellate is found occasionally in the vulva.

In summary, this work could be suitable for publication if it is appropriately scaled. Really it is about the efficiency of T. foetus transit in the bovine repro tract. From the article, I can conclude that it is possible but not very probable (although that data is not shown). The emphasis is what could happen but its unrealistic scenario. As I said, there are virtually zero reports of cows that

become randomly infected. It's a "possibility" but it would be wrong to conclude this is a typical part of the life cycle- at least from the data presented here.

68-80 What is the relevance of saying what all the different trichs do? It seems to be distracting from the article.

83: Remarkably? Perhaps the wrong word choice. Its not remarkable that the feces falls from the rectum and onto the vulva. And is this really "infection"? The reference simply shows the fecal organisms can be found in the female reproductive tract.

104: Where are the reports of verifiable virgin bulls with T foetus?

The finding that T. foetus DNA is in the feces of infected bovines is not surprising. However, it is highly surprising
164: incubated at 19C??

180: Does CFW preferentially stain pseudocysts? Reference?

184: This shows chromatin but 2 nuclei does not mean 2n or n via meiosis

188 nutrient deprivation led to a population with 1 nucleus...? But they are "larger"

216 "more than twice their DNA content".....?

220 I cannot find anywhere that PCN is used with T. foetus.

244 13% viable at pH3 is a remarkable result. So much so that I would challenge if this was possibly miscalculated or mixed.

Also the abomasum has a pH of about 2. Why wasn't that evaluated?

245-253 "sub spherical trophozoites"so pseudocysts? Or are you describing a new stage?

327 in concordance?

330 How can CFW be used to positively identify pseudocysts when it's a "non specific fluorochrome"?

Refer to papers 25 and 26- where is the evidence of infected virgin bulls? Original references please.

Why is it so remarkable for T. foetus to pass through the GI tract, if we already know it can do this in pigs and cats.

335 this is not new, that the pseudocyst could help survive extremes

338 The presence of the pseudocyst form "is significant"??

450: This requires some additional detail about the feces. How much feces, how many cultures? In other words, how much feces had to be examined to find 1 trophozoite?

476 and elsewhere. How many trophs did you use and recover? There are no details.

483: how was the TYM pH adjusted?

Figure 1 shows spherical structures recovered from feces.

They are WGA and dapi positive.....so they are T. foetus? I don't understand that conclusion. Could they be some other protozoa?

What is the procedure for finding these parasites in the feces? Its not given in the methods.

Figure 2: please differentiate between trophs and "spherical structures"

The results section doesn't say this but figure 3 does.....only 1 of 5 animals had PCR positive feces. How many culture positive feces were there? 1?? If they are not PCR positive I have a hard time believing they are all culture positive

Figure 4: In panel A it shows the parasites decay over time in BFE but then in panel B it reports a 2% growth rate. If its declining numbers of parasites/mL it can't have a 2% growth rate. In E there are supposedly difference in pics from TYM and BFE but they are not labeled, nor do they show different numbers of nuclei. Any, the number of nucleoli does not necessarily mean "n"
Can you prove that the objects in these images are not bovine epithelial cells? In E, its simply a cell with a DAPI positive nucleus.

Figure 5: Some of this belongs in the methods. D and E are confusing. These are stained with tubulin. Does that tell you something? A supposed flagella is marked with the arrow. There is nothing convincing about these images that they are T. foetus. Perhaps if they can explain why CFW means T. foetus positive, then maybe I will understand better.

Why isn't there any pictures of the nuclei getting bigger. Nuclei getting 3 times bigger would at least be something that could be shown in a picture whereas most of the pictures here are non diagnostic. In panel G, what are the different groups? All this shows is that there is a dapi positive nucleus and PCNA staining.

Figure 6: you are forgetting the pH of the abomasum @ about 2. In panel B, why aren't the rates for pH 3 shown? Panel C: this might be "representative" but I'm guessing they searched and searched to find a single cell. Any images of concentrated collections of cells or just 1 in the corner? Again, CFW staining is presumed to be a T. foetus. What does E show in terms of significance? That they were mainly dead? The description of panel F does not make sense/cannot be determined.

1x10⁸ parasites were given orally for 5 days! Does anyone ever expect a cow to eat this much material from an aborted fetus? Maybe, but certainly not in feces.

Staff Comments:

Preparing Revision Guidelines

Please return the manuscript within 60 days; if you cannot complete the modification within this time period, please contact me. If you do not wish to modify the manuscript and prefer to submit it to another journal, please notify me of your decision immediately so that the manuscript may be formally withdrawn from consideration by Microbiology Spectrum.

This paper examines transit of *T. foetus* through the GI tract. The authors propose they have discovered a new mechanism whereby cows ingest oocysts, defecate them onto the vulva, and become infected AND possibly perpetuate the infection via sodomy.

The concatenation of experiments is somewhat confusing- in that the majority of the experiments do not show the lifecycle resulting in early embryonic death. Moreover, there are no experiments that show sexual transmission following ingestion.

I think the paper is worthy of publication but it needs a major revision so that it is not over reaching in its conclusions. That is, what this paper really attempts to show, is transit of *T. foetus* through the bovine GI tract. What they do show, is that the parasite can survive in feces and at different pH. They show that in infected herds you can find PCR positive feces, and they show that if you feed cows excessive numbers of trich you can find a few in feces (1 of 5 cows and 1 of 5 bulls). And this is not surprising since the authors begin by talking about how *T. foetus* can transition through the GI system of pigs and of course cats.

Key references: those of infections where there is no sexual reproduction (no bull), etc, are completely lacking. Indeed the only references of connumdrums occur when a fecal flagellate is found occasionally in the vulva.

In summary, this work could be suitable for publication if it is appropriately scaled. Really it is about the efficiency of *T. foetus* transit in the bovine repro tract. From the article, I can conclude that it is possible but not very probable (although that data is not shown). The emphasis is what *could* happen but its unrealistic scenario. As I said, there are virtually zero reports of cows that become randomly infected. It's a "possibility" but it would be wrong to conclude this is a typical part of the life cycle- at least from the data presented here.

68-80 What is the relevance of saying what all the different trichs do? It seems to be distracting from the article.

83: Remarkably? Perhaps the wrong word choice. Its not remarkable that the feces falls from the rectum and onto the vulva. And is this really "infection"? The reference simply shows the fecal organisms can be found in the female reproductive tract.

104: Where are the reports of verifiable virgin bulls with *T. foetus*?

The finding that *T. foetus* DNA is in the feces of infected bovines is not surprising. However, it is highly surprising

164: incubated at 19C??

180: Does CFW preferentially stain pseudocysts? Reference?

184: This shows chromatin but 2 nuclei does not mean 2n or n via meiosis

188 nutrient deprivation led to a population with 1 nucleus...? But they are "larger"

216 "more than twice their DNA content".....?

220 I cannot find anywhere that PCN is used with T. foetus.

244 13% viable at pH3 is a remarkable result. So much so that I would challenge if this was possibly miscalculated or mixed. Also the abomasum has a pH of about 2. Why wasn't that evaluated?

245-253 "sub spherical trophozoites"so pseudocysts? Or are you describing a new stage?

327 in concordance?

330 How can CFW be used to positively identify pseudocysts when it's a "non specific fluorochrome"?

Refer to papers 25 and 26- where is the evidence of infected virgin bulls? Original references please.

Why is it so remarkable for T. foetus to pass through the GI tract, if we already know it can do this in pigs and cats.

335 this is not new, that the pseudocyst could help survive extremes

338 The presence of the pseudocyst form "is significant"??

450: This requires some additional detail about the feces. How much feces, how many cultures? In other words, how much feces had to be examined to find 1 trophozoite?

476 and elsewhere. How many trophs did you use and recover? There are no details.

483: how was the TYM pH adjusted?

Figure 1 shows spherical structures recovered from feces.

They are WGA and dapi positive.....so they are T. foetus? I don't understand that conclusion. Could they be some other protozoa?

What is the procedure for finding these parasites in the feces? Its not given in the methods.

Figure 2: please differentiate between trophs and "spherical structures"

The results section doesn't say this but figure 3 does.....only 1 of 5 animals had **PCR** positive feces. How many culture positive feces were there? 1?? If they are not PCR positive I have a hard time believing they are all culture positive

Figure 4: In panel A it shows the parasites decay over time in BFE but then in panel B it reports a 2% growth rate. If its declining numbers of parasites/mL it can't have a 2% growth rate. In E there are

supposedly difference in pics from TYM and BFE but they are not labeled, nor do they show different numbers of nuclei. Any, the number of nucleoli does not necessarily mean “n”

Can you prove that the objects in these images are not bovine epithelial cells? In E, its simply a cell with a DAPI positive nucleus.

Figure 5: Some of this belongs in the methods. D and E are confusing. These are stained with tubulin. Does that tell you something? A supposed flagella is marked with the arrow. There is nothing convincing about these images that they are T. foetus. Perhaps if they can explain why CFW means T. foetus positive, then maybe I will understand better.

Why isn't there any pictures of the nuclei getting bigger. Nuclei getting 3 times bigger would at least be something that could be shown in a picture whereas most of the pictures here are non diagnostic. In panel G, what are the different groups? All this shows is that there is a dapi positive nucleus and PCNA staining.

Figure 6: you are forgetting the pH of the abomasum @ about 2. In panel B, why aren't the rates for pH 3 shown? Panel C: this might be “representative” but I'm guessing they searched and searched to find a single cell. Any images of concentrated collections of cells or just 1 in the corner? Again, CFW staining is presumed to be a T. foetus. What does E show in terms of significance? That they were mainly dead? The description of panel F does not make sense/cannot be determined.

1×10^8 parasites were given orally for 5 days! Does anyone ever expect a cow to eat this much material from an aborted fetus? Maybe, but certainly not in feces.

Initially, we would like to thank the Reviewers and Editor for their valuable time and comments which definitely contributed to improve the manuscript. We have appreciated their compliments, suggestions, and criticism on the quality, strengths and limitations of our main findings and writing. We would also like to thank the opportunity to answer the questions and concerns about our manuscript.

We have accepted the points raised by the referees and the alterations are highlighted in blue in the revised manuscript. Thank you for your efforts on our behalf, and we hope to hear from you at your earliest convenience.

Reviewer #1:

In this manuscript the authors investigate the potential role of the gut as a site of infection for *Tritrichomonas foetus* in bovine. The author combine experimental infections experiments with survey of naturally infected bovine to demonstrate the gut-UGT infections can take place in addition to STI route of infection. This is a very interesting and important paper that resolves an important conundrum of some infection pattern among bovine that could not be simply explained by STI transmissions. It is also complementing existing data for oral-fecal route transmission of the same parasite in other hosts (cats, dogs, pigs). I have only a few points to improve this overall very good manuscript.

Main points

-It would have been important to sequence the PCR products detected from natural infections and compare the sequences with published sequences. As there is plenty of PCR products, and if they are still available, this should be straightforward to generate. How heterogenous are these sequences and how do they compare with those from various isolates (e.g. cats, dogs, pigs), see for example Kleina et al. (2004) and related papers. Kleina et al. (2004). Molecular phylogeny of Trichomonadidae family inferred from ITS-1, 5.8S rRNA and ITS-2 sequences. International Journal for Parasitology, 34: 963-970.

Response: We appreciate the reviewer's comment, and we added Figure 4 with the requested data.

-The sentence "Unlike other parasitic protozoa, *T. foetus* has only one morphological stage, the trophozoite." Is not accurate. Indeed, it is contradicted in the following sentence that is referring to the pseudocyst form. It would be more accurate, and more straightforward, to simply refer to trophozoite and pseudocysts as the two cellular forms known for that parasite. The related sentence in the discussion needs to be edited accordingly (section starting at line 320).

Response: Thanks for the observation; we modified the sentences.

-Similarly, the sentence "Thus, these organisms are frequently found in the samples obtained from bulls when analyzed by the standard diagnostic methods." Should be made more precise as the frequency of such gut derived infections are context dependent and have only rarely been detected in some contexts, such as in Europe. See for instance Bailey et al. (2021). Bailey et al., (2021). Sporadic isolation of *Tetratrichomonas* species from the cattle urogenital tract.

Parasitology, 146: 1109-1115. Note that in this paper that no *T. foetus* was detected in the gut metatranscriptomics data from another study.

Response: Thanks for the observation; we modified the sentences. On the other hand, we consider that probably *T. foetus* has not been detected in metatranscriptomic analyses due to its occasional presence in the gastrointestinal tract.

-When discussing the pseudocyst it is important to consider the observations by Pereira-Neves et al. (2012) that the pseudocyst were even more cytotoxic than the trophozoites. This could be relevant in the transmission from gut to the UGT, a transmission route suggested in this study. Pereira-Neves A, Nascimento LF, Benchimol M (2012) Cyto-toxic effects exerted by *Tritrichomonas foetus* pseudocysts. *Protist*, 163:529-543.

Response: Thanks for the observation. Now, we have improved the discussion in this context.

Other points

-*T. foetus* was also isolated from a few human patients, see review by Martiz et al. (2014) and more recently see paper by Zuzuki et al. (2016). Maritz et al. (2014). What is the importance of zoonotic Trichomonads for human health? *Trends in Parasitology*, 30: 333-341.

Zuzuki et al. (2016). Characterization of a human isolate of *Tritrichomonas foetus* (cattle/swine genotype) infected by a zoonotic opportunistic infection. *Journal of Veterinary Medical Science* 78: 633-640.

Response: We appreciated the reviewer's comment. Now, we have added this information to the discussion.

-pH3 conditions -> pH 3 conditions

Response: Thanks for the observation; we have corrected the mistake in the text.

Reviewer #2:

This manuscript authored by Martinez CI and colleagues is addressing possibility that *Tritrichomonas foetus*, a trichomonad protozoa can be transmitted by fecal contamination of the urogenital tract of female cattle followed by sexual transmission. They had logically showed: first, that the trichomonad parasite survived and passed through the gastrointestinal tract, and was able to reach the urogenital tract of cows by fecal contamination following oral inoculation of trophozoites; second, the trophozoites of *T. foetus* was found in the fecal materials of naturally infected cattle; third, these trophozoites survived days in feces; fourth; they remained viable in water for up to 72 hours; last, but not the least, they tolerated low pH conditions mimicking the gastrointestinal tract of cattle. All these data were solidly laid out. Based on these data, they proposed that trophozoites that originate in the gastrointestinal tract and contaminate the urogenital tract of cows serve as a new spread route for the parasites to be sexually transmitted among cattle herds.

Whether this newly proposed route might be true needs to be confirmed. Nevertheless, some already well-known facts seem contradicting, or at least not supporting such a notion. First, the methods for prevention and control of bovine trichomonosis solely targeting the sexual

transmission route have achieved eradication of the disease on state or even country wide scale. Many European countries have achieved it by widespread use of artificial insemination (AI) (<https://www.microbiologyresearch.org/content/journal/jmm/10.1099/jmm.0.047365-0#tab2>). Wyoming State of the United State America has reached eradication by testing bulls and culling T. foetus-positive bull (<https://veterinaryresearch.biomedcentral.com/articles/10.1186/s13567-021-00996-w>). These successful achievements undoubtedly pinpoint the sexual contact as the main, if not the only transmission route of the disease. Second, five cows were tested for *T. foetus* trophozoites in both the gastrointestinal and urogenital tracts. One animal was tested positive in both tracts, whereas the other two were only positive in the latter, which would be hard to be explained by fecal contamination. They seemed to be infected by sexual activities. To confirm the newly proposed route of transmission, an isolated herd of virgin heifer with infection of the parasites in the gastrointestinal tract is needed. These females are allowed to mate with *T. foetus*-negative bulls, and female heifers become positive after mating in the absence of the parasite in their feces. This reviewer understands this is very challenging to achieve. Without such confirmative data your proposed notion of new transmission route is hard to believe due to the reason just mentioned. Even this route is proven to be true, it may just play a very minor role in some herds or no role at all in others. Therefore, it is of no great effect on the current methods for control and prevention.

Response: We appreciate these comments and have made modifications to the manuscript considering the reviewer's observations. We modified the results description and the general conclusions. In this context, we consider that while this possible form of transmission is less common, it could be important in regions where extensive livestock systems prevail, coupled with specific factors such as climatic conditions during the breeding season (rainfall, humidity, etc.), low-lying areas that promote water accumulation (water puddles), and young pastures coinciding with the breeding season (which can affect fecal consistency). During the present study, we captured the following photographs of situations observed in different extensive systems analyzed in Buenos Aires (Argentina):

On the other hand, in future analyses, we should take into account certain characteristics of the parasite's biology that are currently not being considered but could be relevant for evaluating possible alternative transmission forms of *T. foetus*. These characteristics include: **a)** the ability of *T. foetus* to form cyst-like structures under stress conditions. **b)** *T. foetus* is capable of endoreplicating its DNA and generating multiple nuclei within a single parasite under certain stress conditions*. **c)** These multinucleate parasites can give rise to numerous parasites through multiple

fissions when optimal external conditions are restored*. Interestingly, in the analysis performed by Iriarte et al.*, we have quantified up to 16 nuclei in a single parasite:

* DOI: [10.1128/spectrum.03251-22](https://doi.org/10.1128/spectrum.03251-22)

-You only mentioned the direct economic losses due to abortion and infertility. However, there are indirect losses as well such as testing fees and veterinarian costs. See the paper - <https://veterinaryresearch.biomedcentral.com/articles/10.1186/s13567-021-00996-w>

Response: Thanks for the observation; we modified the sentences.

-"*T. foetus* is also a commensal parasite found in the intestines and stomachs of pigs." This statement is not correct. A commensal CAN'T be a parasite, or vice versa by definition. They belong to very different categories.

Response: Thanks for the observation; we have corrected the mistake in the text.

-A scale bar is needed for all micrographs in figures.

Response: We appreciated the reviewer's comment. In the present manuscript, we have now added the scale bar.

-Figure 2C: It is not clear whether the samples were from same cow or different cows.

Response: Thanks; we have improved the description in the manuscript.

-Figure 4A: In M&M: growth rate = $[\ln CC(t) - \ln CC(0)] / (t - 0)$, where $CC(t)$ and $CC(0)$ are the parasites counts per milliliter at time t and time zero, respectively, and t is the time of incubation (45)". This figure shows parasites/ml. Please explain the discrepancy.

Response: The growth curve and the growth rate are different concepts in the study of microbial growth. The growth curve (Fig. 5A) is a graphical representation that shows how the population of microorganisms changes over time. The growth rate (Fig. 5B) refers to the speed or rate at which a population of microorganisms increases in number during a specific time interval. The growth rate depends on various factors, including nutrient availability, environmental conditions, and the microorganism's ability to reproduce and divide.

Figure 4E: Labels "TYM and BFE: are missing in the panels.

Response: Thanks for the observation, we have modified the figure description in the manuscript.

-For equipment and reagents, please list providers and their location including city & country.

Response: Thanks for the observation; we included the information in the present manuscript.

-You described two ways to apply *T. foetus* to a cover slip for downstream application such as staining, one is direct; another is culturing trophozoites for hours. In culture, no medium was specified. Why did the trophozoites need to be cultured on a coverslip prior to staining etc.?

Response: The parasites are incubated for hours in a culture medium on a coverslip in order to increase the number of adhered parasites in order to conduct quantification assays (e.g., counting nuclei) or during an IFA assay, in which many parasites are lost during successive washings.

In other cases, when the presence of some of the structures of interest in the samples is suspected to be low, the samples are centrifuged and resuspended in a small volume to concentrate such samples onto the coverslip (for example, staining cysts with CFW).

-How were the trophozoites in feces and mucus samples quantified as mentioned in L409-410?

Response: We have modified the manuscript considering this observation.

-PCR primer concentration should not be 1 mM. It is a typo of 1 μ M, isn't it?

Response: Thanks for the comment, we have corrected the mistake in the manuscript.

-Only two cows were used in the experiments of gastrointestinal infection. The sample size is too small. How was the infection performed? What treatment was used to clear the infections?

Response: We conducted the experiment with two animals, as that was the approved total animal number by the Ethics Committee. It's important to note that these two animals were already earmarked for culling due to reasons unrelated to health. The assay was conducted in the experimental area of INTECH-Chascomús, where there are other animals involved in different experimental studies. We utilized a suitable infrastructure separated from other animal use areas and restricted access to that space throughout the reproductive season. After the experiment was over, we conducted inspections in the area occupied by the animals during the assay, performing superficial soil movements to promote the desiccation of animal feces, which is lethal to parasites. Lastly, the use of that area with animals was not authorized until the next reproductive period.

-References: the names of genus and species should be italicized. Also, the font size and line space should be the same as those of the main text.

Response: Thanks for the observation; we modified the manuscript.

-L265: "microscopic examination followed by cell culture" The order seems reversed. Usually cell culture takes place first, followed by microscopy.

Response: Thanks; we modified the phrase in the text.

-L267-268: AI should be mentioned as well since it is the most effective way to control and eradication of bovine trichomonosis, which has been proven in most European countries.

Response: Thanks for the comment; we modified the sentences.

-L467-468: In what a volume were 1×10^5 parasites (*T. foetus* K strain) inoculated in the supernatants?

Response: Thanks for the observation; we improved the protocol description.

Minor issues:

Be consistent, both tritrichomonosis and trichomonosis are used for the same disease caused by *T. foetus*.

Response: Thanks for this observation.

The word "eliminate" seems not a right word to describe trophozoites in host feces, discharge or pass or secrete?

Response: We have modified the word in all sentences.

-L17: "the life cycle of the protozoan *T. foetus*"

Response: We have corrected the mistake in the manuscript.

-L43-44: "Unlike other parasitic protozoa, *T. foetus* has only one morphological stage, the trophozoite". The statement is inaccurate since quite a few species of protozoa have only trophozoite stage.

Response: Thanks for the comment; we modified the sentence.

-"*T. foetus* in samples from virgin bulls or the presence of bull negative samples and cow positive samples in the same bovine herds are clearly situations difficult to explain based on the current knowledge of *T. foetus* life cycle." The exact same sentences appear multiple times in the manuscript. Please reword them differently.

Response: We modified the sentences.

We modified the next phrases and mistakes:

L76: "amphibian intestinal parasite,"

L77: "mammal intestine"

L273: "some trichomonads such"

L326: "*T. muris* fecal-oral transmission"

L345: "multiplication by multiple fission"

L572: "an alpha = 0.05."

Reviewer #3:

This paper examines transit of *T. foetus* through the GI tract. The authors propose they have discovered a new mechanism whereby cows ingest oocysts, defecate them onto the vulva, and become infected AND possibly perpetuate the infection via sodomy.

The concatenation of experiments is somewhat confusing- in that the majority of the experiments do not show the lifecycle resulting in early embryonic death. Moreover, there are no experiments that show sexual transmission following ingestion.

I think the paper is worthy of publication but it needs a major revision so that it is not overreaching in its conclusions. That is, what this paper really attempts to show, is transit of *T. foetus* through the bovine GI tract. What they do show, is that the parasite can survive in feces and at different pH. They show that in infected herds you can find PCR positive feces, and they show that if you feed cows excessive numbers of trich you can find a few in feces (1 of 5 cows and 1 of 5 bulls). And this is not surprising since the authors begin by talking about how *T. foetus* can transition through the GI system of pigs and of course cats.

Key references: those of infections where there is no sexual reproduction (no bull), etc, are completely lacking. Indeed the only references of connumdrums occur when a fecal flagellate is found occasionally in the vulva.

Response: Thanks for the observation; we modified the manuscript.

-In summary, this work could be suitable for publication if it is appropriately scaled. Really it is about the efficiency of *T. foetus* transit in the bovine repro tract. From the article, I can conclude that it is possible but not very probable (although that data is not shown). The emphasis is what could happen but its unrealistic scenario. As I said, there are virtually zero reports of cows that become randomly infected. It's a "possibility" but it would be wrong to conclude this is a typical part of the life cycle- at least from the data presented here.

Respuesta: We truly appreciate the reviewer for pointing this out. Now, we have changed the manuscript in general.

-68-80 What is the relevance of saying what all the different trichs do? It seems to be distracting from the article.

Response: We considered that a short description of the different trichomonads and the extensive range of hosts and body sites in which they are found would enrich the manuscript.

-83: Remarkably? Perhaps the wrong word choice. It's not remarkable that the feces falls from the rectum and onto the vulva. And is this really "infection"? The reference simply shows the fecal organisms can be found in the female reproductive tract.

Response: Thanks for the observation; we have modified the sentence.

-104: Where are the reports of verifiable virgin bulls with *T foetus*?

Response: We really thank for this reviewer observation. We added the reference.

-The finding that *T. foetus* DNA is in the feces of infected bovines is not surprising. However, it is highly surprising

Response: At this point, we highlight that it was essential to keep the samples for several days in a culture medium to increase the number of parasites, which facilitated their detection by PCR.

-164: incubated at 19C??

Response: Thanks; we have modified the text.

-180: Does CFW preferentially stain pseudocysts? Reference?

Response: This is an important point raised by the reviewer, and we truly appreciate that. Calcofluor White can be used to stain and detect cysts or pseudocysts of certain parasites, particularly those that have chitin or cellulose in their cyst walls. In pseudocyst forms, the staining is usually more diffuse, and in true cysts, the existence of the cystic wall can be clearly observed. In relation to this, we have improved the description in the manuscript and added the references (References: DOI:10.1016/j.protis.2010.03.006, DOI:10.3389/fcimb.2019.00430)

-184: This shows chromatin but 2 nuclei does not mean 2n or n via meiosis

Response: We improved the sentence and the figure legend. In figure 4D (Fig. 5D in the last version of the manuscript): 1N, 2N, and >2N indicate nuclei number, not DNA content (N=nuclei).

-188 nutrient deprivation led to a population with 1 nucleus...? But they are "larger"

Response: Thanks; we have modified the manuscript.

-216 "more than twice their DNA content".....?

Response: Thanks; we have modified the sentence.

-220 I cannot find anywhere that PCN is used with *T. foetus*.

Response: We really appreciate this reviewer's observation. We added the reference to the manuscript.

-244 13% viable at pH3 is a remarkable result. So much so that I would challenge if this was possibly miscalculated or mixed. Also the abomasum has a pH of about 2. Why wasn't that evaluated?

Response: Thank you very much for this observation. In this version of the manuscript, we have included the experiment conducted at pH 2. In this context, we emphasize that in this experiment, the parasites were previously pre-incubated for 24 hours in water (stress condition) and then incubated in culture media at pH 2. We performed this experimental setup considering that if the parasites come from the external environment (upon oral ingestion), they were previously subjected to some form of stress (e.g., nutritional).

-245-253 "sub spherical trophozoites"so pseudocysts? Or are you describing a new stage?

Response: We truly appreciate your comments because this observation allowed us to improve the description of these structures in the manuscript.

-327 in concordance?

Response: Thanks for the observation; we modified the phrase.

-330 How can CFW be used to positively identify pseudocysts when it's a "non specific fluorochrome"?

Response: The non-specificity of calcofluor white refers to its ability to bind to different types of polysaccharides rather than specifically targeting a particular molecule or structure. It has a broad affinity for cellulose, chitin, and other β -1,3- and β -1,4-linked polysaccharides, which are commonly found in the cell walls of many organisms, including fungi, algae, and some bacteria. Calcofluor White can be used to stain and detect cysts of certain parasites, particularly those that have chitin or cellulose in their cyst walls. Some examples of cyst-forming parasites that can be visualized using Calcofluor White include: *Giardia lamblia*, *Cryptosporidium spp* and *Acanthamoeba spp*.

-Refer to papers 25 and 26- where is the evidence of infected virgin bulls? Original references please.

Response: We added the references to the manuscript.

-Why is it so remarkable for *T. foetus* to pass through the GI tract, if we already know it can do this in pigs and cats.

Response: We have highlighted this characteristic in particular, considering that the ruminant gastrointestinal tract has greater structural and functional complexity.

-335 this is not new, that the pseudocyst could help survive extremes

Response: We have modified the manuscript regarding the description of these structures. Moreover, we consider that it is necessary to analyze in detail the specific role of pseudocysts and cyst-like structures in the future in terms of their resistance to extreme conditions.

-338 The presence of the pseudocyst form "is significant"??

Response: Thanks for the observation; we modified the phrase.

-450: This requires some additional detail about the feces. How much feces, how many cultures? In other words, how much feces had to be examined to find 1 trophozoite?

Response: Thanks for the observation; we have included this information in the manuscript.

-476 and elsewhere. How many trophs did you use and recover? There are no details.

Response: Thanks; we have added this information to the manuscript.

-483: how was the TYM pH adjusted?

Response: Thanks; we have added this information to the manuscript.

-Figure 1 shows spherical structures recovered from feces. They are WGA and dapi positive.....so they are *T. foetus*? I don't understand that conclusion. Could they be some other protozoa?. What is the procedure for finding these parasites in the feces? Its not given in the methods.

Response: During the sampling of feces and mucus, we took an aliquot and carried out successive washings with PBS and staining with WGA. After incubating the samples in culture medium for

trichomonas, these structures gradually disappeared, and the number of motile trophozoites increased. We believe that we were able to easily observe these structures throughout the samplings due to the number of parasites used for the experimental oral infection. On the other hand, we have justified with references in this manuscript the use of WGS for the staining of resistance structures.

-Figure 2: please differentiate between trophs and "spherical structures"

Response: We have improved the writing of the manuscript in this regard.

-The results section doesn't say this but figure 3 does.....only 1 of 5 animals had PCR positive feces. How many culture positive feces were there? 1?? If they are not PCR positive I have a hard time believing they are all culture positive

Response: We have improved the description in the manuscript, and in this new version of it, we have sequenced the PCR fragments obtained from animal samples (mucus and feces).

-Figure 4: In panel A it shows the parasites decay over time in BFE but then in panel B it reports a 2% growth rate. If its declining numbers of parasites/mL it can't have a 2% growth rate.

Response: The growth curve and the growth rate are different concepts in the study of microbial growth. The growth curve (Fig. 4A) is a graphical representation that shows how the population of microorganisms changes over time. The growth rate (Fig. 4B) refers to the speed or rate at which a population of microorganisms increases in number during a specific time interval. The growth rate depends on various factors, including nutrient availability, environmental conditions, and the microorganism's ability to reproduce and divide.

In E there are supposedly difference in pics from TYM and BFE but they are not labeled, nor do they show different numbers of nuclei. Any, the number of nucleoli does not necessarily mean "n"

Response: We have improved the writing of the manuscript.

-Can you prove that the objects in these images are not bovine epithelial cells? In E, its simply a cell with a DAPI positive nucleus.

Response: Parasites are routinely grown in 8–15 ml tubes and passaged daily. In this assay, we take aliquots from the supernatants where only *T. foetus* (trophozoites) is found, as it can remain in suspension due to the movement of its flagella. If there were epithelial cells in the sample, they would be at the bottom of the tube.

-Figure 5: Some of this belongs in the methods. D and E are confusing. These are stained with tubulin. Does that tell you something? A supposed flagella is marked with the arrow. There is nothing convincing about these images that they are *T. foetus*. Perhaps if they can explain why CFW means *T. foetus* positive, then maybe I will understand better.

Response: Thanks for this observation; we have improved the writing of the manuscript.

-Why isn't there any pictures of the nuclei getting bigger. Nuclei getting 3 times bigger would at least be something that could be shown in a picture whereas most of the pictures here are non

diagnostic. In panel G, what are the different groups? All this shows is that there is a dapi positive nucleus and PCNA staining.

Response: We have taken this observation into account and have modified the description of this figure in the manuscript.

-Figure 6: you are forgetting the pH of the abomasum @ about 2. In panel B, why aren't the rates for pH 3 shown? Panel C: this might be "representative" but I'm guessing they searched and searched to find a single cell. Any images of concentrated collections of cells or just 1 in the corner? Again, CFW staining is presumed to be a *T. foetus*. What does E show in terms of significance? That they were mainly dead? The description of panel F does not make sense/cannot be determined.

Response: In this version of the manuscript, we have included the experiment at pH2. We consider that for the pH2 analysis, it was more important to analyze viability than growth rate, taking into account that the parasites would take 3 hours to go through this adverse pH condition (a short time to replicate). We also consider it more interesting to analyze the subsequent growth when optimal pH conditions are restored. We have eliminated panel C and justified the use of Calcofluor white throughout the manuscript. We have changed the figure and added information to the manuscript.

- 1×10^8 parasites were given orally for 5 days! Does anyone ever expect a cow to eat this much material from an aborted fetus? Maybe, but certainly not in feces.

Response: We agree with the reviewer's comment regarding the excessive number of parasites administered. We used that quantity of parasites considering that the Ethics Committee only allowed us to use two animals for the assay. Then our main objective was to demonstrate that the parasites could traverse the gastrointestinal tract and be eliminated alive with feces. In this context, it is interesting to note that for the experimental orogastric infection in cats, 2×10^6 parasites were used (DOI: 10.2460/ajvr.2001.62.1690).

On the other hand, it is true that a cow could never ingest this quantity of parasites in nature, however in future analyses, we should take into account certain characteristics of the parasite's biology that are currently not being considered but could be relevant for evaluating possible alternative transmission forms of *T. foetus*. These characteristics include: a) the ability of *T. foetus* to form cyst-like structures under stress conditions; b) *T. foetus* is capable of endoreplicating its DNA and generating multiple nuclei within a single parasite under certain stress conditions*. c) These multinucleate parasites can give rise to numerous parasites through multiple fissions when optimal external conditions are restored*. Interestingly, in the analysis performed by Iriarte et al.*, we have quantified up to 16 nuclei in a single parasite:

* DOI: [10.1128/spectrum.03251-22](https://doi.org/10.1128/spectrum.03251-22)

August 7, 2023

Dr. Veronica Mabel Coceres
Instituto Tecnológico de Chascomus
Laboratorio de Parásitos Anaerobios
Intendente Marino km 8.2
Chascomús, Buenos Aires 7130
Argentina

Re: Spectrum00429-23R1 (*Tritrichomonas foetus* life cycle: report of a new spread route in bovines)

Dear Dr. Veronica Mabel Coceres:

The review of your manuscript is now almost complete, but I would be grateful if you could address the few remaining minor issues that the Reviewers have highlighted. In separate comments directly to me as Editor, one of the reviewers felt strongly that there was insufficient evidence to support "a new route of infection", hence the suggested title/focus change (which the reviewer did consider interesting and worthy of acceptance). I would be grateful if you could give this some consideration when preparing your revised manuscript.

Thank you for submitting your manuscript to Microbiology Spectrum. As you will see your paper is very close to acceptance. Please modify the manuscript along the lines I have recommended. As these revisions are quite minor, I expect that you should be able to turn in the revised paper in less than 30 days, if not sooner. If your manuscript was reviewed, you will find the reviewers' comments below.

When submitting the revised version of your paper, please provide (1) point-by-point responses to the issues raised by the reviewers as file type "Response to Reviewers," not in your cover letter, and (2) a PDF file that indicates the changes from the original submission (by highlighting or underlining the changes) as file type "Marked Up Manuscript - For Review Only". Please use this link to submit your revised manuscript. Detailed instructions on submitting your revised paper are below.

Link Not Available

Sincerely,

Neil Mabbott

Reviewer comments:

Reviewer #1 (Comments for the Author):

The authors have addressed all key issues identified in the earlier version of the manuscript. This has clearly improved the manuscript.

I would add one comment relating to the one of the other reviewers' comment:

-"T. foetus is also a commensal parasite found in the intestines and stomachs of pigs." This statement is not correct. A commensal CAN'T be a parasite, or vice versa by definition. They belong to very different categories.

I disagree with this narrow perspective. *T. foetus* should be considered as a symbiont (as originally defined by du Barry). And, importantly, depending on the nature of the interaction with its host, which is context dependent (host immunological status, microbiota status/composition/activity, diet etc...), can be a commensal or a parasite, and potentially, a mutualists. To my mind this a more accurate perspective on the biology of this organism. But I agree that the sentence needs to be made more precise.

Reviewer #2 (Comments for the Author):

The authors have adequately addressed this reviewer's concerns during revision.

Reviewer #3 (Comments for the Author):

I reviewed this previously. There are NO "virgin animals" infected like the authors claim in reference 11

In summary, this paper should be "Prolonged survival of T. foetus in the GI tract, bovine fecal extract and water"

Some of it might be a language editing issue.

Why does the CFW prove everything is a T. foetus? Other things are CFW positive.

I think the authors are correct that some trichomonads might be spread in the feces of bovines but I don't think that cattle get venereal infections. I disagree with that conclusion.

Again, I think the paper should indicate the survival in water and feces and the authors can debate about cyst or not, but it's not a route of infection by a parasite. Unless they are saying this is a different trichomonad that is spread by a fecal oral route like T suis

Preparing Revision Guidelines

Please return the manuscript within 60 days; if you cannot complete the modification within this time period, please contact me. If you do not wish to modify the manuscript and prefer to submit it to another journal, please notify me of your decision immediately so that the manuscript may be formally withdrawn from consideration by Microbiology Spectrum.

We'd like to express our gratitude again to the Reviewers and Editor for their remarks, which helped us enhance the manuscript. The suggestions of the referees have been followed, and the modifications are highlighted in blue in the revised version.

Reviewer #1:

The authors have addressed all key issues identified in the earlier version of the manuscript. This has clearly improved the manuscript.

I would add one comment relating to the one of the other reviewers' comment: "*T. foetus* is also a commensal parasite found in the intestines and stomachs of pigs." This statement is not correct. A commensal CAN'T be a parasite, or vice versa by definition. They belong to very different categories.

I disagree with this narrow perspective. *T. foetus* should be considered as a symbiont (as originally defined by du Barry). And, importantly, depending on the nature of the interaction with its host, which is context dependent (host immunological status, microbiota status/composition/activity, diet etc...), can be a commensal or a parasite, and potentially, a mutualists. To my mind this a more accurate perspective on the biology of this organism. But I agree that the sentence needs to be made more precise.

Response: Thanks for the observation. Now we have improved the sentences related to this.

Reviewer #2:

The authors have adequately addressed this reviewer's concerns during revision.

Reviewer #3:

I reviewed this previously. There are NO "virgin animals" infected like the authors claim in reference 11

Response: We consider this suggestion, and we have modified the text.

In summary, this paper should be "Prolonged survival of *T. foetus* in the GI tract, bovine fecal extract and water". Some of it might be a language editing issue.

Response: We really appreciate this suggestion, and in this version of the manuscript, we have replaced the title and changed the focus of the conclusions.

Why does the CFW prove everything is a *T. foetus*? Other things are CFW positive.

Response: At this point, it is important to consider that the use of CFW stain was described in general for microorganisms that do not have flagella (fungi, Entamoeba, Cryptosporidium, and Acanthamoeba). In this sense, even though *Giardia lamblia* is an intestinal and flagellated protozoan and forms cysts, the shape of these cysts is different (ovoid), and CFW stain is not a good cyst marker

because the cyst wall is not composed largely of chitin (DOI: 10.1016/0166-6851(89)90063-7). Taking this into account, in experiments with feces or feces extracts (unique assays in this work in which cysts of other microorganisms could have appeared), we performed an IFA assay using the anti-tubulin antibody to demonstrate the internalization of the flagella (as a control).

I think the authors are correct that some trichomonads might be spread in the feces of bovines but I don't think that cattle get venereal infections. I disagree with that conclusion.

Response: We appreciate this suggestion, and in this version of the manuscript, we have changed the focus of the conclusions.

Again, I think the paper should indicate the survival in water and feces and the authors can debate about cyst or not, but it's not a route of infection by a parasite. Unless they are saying this is a different trichomonad that is spread by a fecal oral route like *T. suis*.

Response: Thanks for the observation. Now, we have changed the manuscript in general.

August 15, 2023

Dr. Veronica Mabel Coceres
Instituto Tecnológico de Chascomus
Laboratorio de Parásitos Anaerobios
Intendente Marino km 8.2
Chascomús, Buenos Aires 7130
Argentina

Re: Spectrum00429-23R2 (*Tritrichomonas foetus* life cycle: report of a new spread route in bovines)

Dear Dr. Veronica Mabel Coceres:

Thank you for submitting your manuscript to Microbiology Spectrum in response the suggestions requested by the reviewers and myself as editor.

However, when viewing your submitted files I noted that whereas the marked copy of the manuscript contains the necessary revisions, the "clean" and merged (non-marked) submitted copies of the manuscript etc. do not match or contain the required revisions. This is important since the clean version of the text do not include the revisions removing the emphasis on a new route of transmission etc. as requested after the last round of peer review. The manuscript title and abstract also haven't been updated in the submission system. I would be grateful if you could supply the correctly updated files and update the submission. I cannot make a decision on your manuscript until the correct files and information are supplied.

Link Not Available

Sincerely,

Neil Mabbott

Journals Department
Reviewer comments:

Staff Comments:

Preparing Revision Guidelines

Please return the manuscript within 60 days; if you cannot complete the modification within this time period, please contact me. If you do not wish to modify the manuscript and prefer to submit it to another journal, please notify me of your decision immediately so that the manuscript may be formally withdrawn from consideration by Microbiology Spectrum.

We'd like to express our gratitude again to the Reviewers and Editor for their remarks, which helped us enhance the manuscript. The suggestions of the referees have been followed, and the modifications are highlighted in blue in the revised version.

Reviewer #1:

The authors have addressed all key issues identified in the earlier version of the manuscript. This has clearly improved the manuscript.

I would add one comment relating to the one of the other reviewers' comment: "*T. foetus* is also a commensal parasite found in the intestines and stomachs of pigs." This statement is not correct. A commensal CAN'T be a parasite, or vice versa by definition. They belong to very different categories.

I disagree with this narrow perspective. *T. foetus* should be considered as a symbiont (as originally defined by du Barry). And, importantly, depending on the nature of the interaction with its host, which is context dependent (host immunological status, microbiota status/composition/activity, diet etc...), can be a commensal or a parasite, and potentially, a mutualists. To my mind this a more accurate perspective on the biology of this organism. But I agree that the sentence needs to be made more precise.

Response: Thanks for the observation. Now we have improved the sentences related to this.

Reviewer #2:

The authors have adequately addressed this reviewer's concerns during revision.

Reviewer #3:

I reviewed this previously. There are NO "virgin animals" infected like the authors claim in reference 11

Response: We consider this suggestion, and we have modified the text.

In summary, this paper should be "Prolonged survival of *T. foetus* in the GI tract, bovine fecal extract and water". Some of it might be a language editing issue.

Response: We really appreciate this suggestion, and in this version of the manuscript, we have replaced the title and changed the focus of the conclusions.

Why does the CFW prove everything is a *T. foetus*? Other things are CFW positive.

Response: At this point, it is important to consider that the use of CFW stain was described in general for microorganisms that do not have flagella (fungi, Entamoeba, Cryptosporidium, and Acanthamoeba). In this sense, even though *Giardia lamblia* is an intestinal and flagellated protozoan and forms cysts, the shape of these cysts is different (ovoid), and CFW stain is not a good cyst marker

because the cyst wall is not composed largely of chitin (DOI: 10.1016/0166-6851(89)90063-7). Taking this into account, in experiments with feces or feces extracts (unique assays in this work in which cysts of other microorganisms could have appeared), we performed an IFA assay using the anti-tubulin antibody to demonstrate the internalization of the flagella (as a control).

I think the authors are correct that some trichomonads might be spread in the feces of bovines but I don't think that cattle get venereal infections. I disagree with that conclusion.

Response: We appreciate this suggestion, and in this version of the manuscript, we have changed the focus of the conclusions.

Again, I think the paper should indicate the survival in water and feces and the authors can debate about cyst or not, but it's not a route of infection by a parasite. Unless they are saying this is a different trichomonad that is spread by a fecal oral route like *T. suis*.

Response: Thanks for the observation. Now, we have changed the manuscript in general.

August 16, 2023

Dr. Veronica Mabel Coceres
Instituto Tecnológico de Chascomus
Laboratorio de Parásitos Anaerobios
Intendente Marino km 8.2
Chascomús, Buenos Aires 7130
Argentina

Re: Spectrum00429-23R3 (Prolonged survival of venereal *Tritrichomonas foetus* parasite in the gastrointestinal tract, bovine fecal extract, and water)

Dear Dr. Veronica Mabel Coceres:

Your manuscript has been accepted, and I am forwarding it to the ASM Journals Department for publication. You will be notified when your proofs are ready to be viewed.

Sincerely,

Neil Mabbott
Editor, Microbiology Spectrum
